# Biodesulfurization Induces Reprogramming of Sulfur Metabolism in *Rhodococcus qingshengii* IGTS8: Proteomics and Untargeted Metabolomics

Aurélie Hirschler,[a] Christine Carapito,[a] Loïc Maurer,[b,c] Julie Zumsteg,[b] Claire Villette,[b] Dimitri Heintz,[b] Christiane Dahl,[d] Ashraf Al-Nayal,[e] Vartul Sangal,[f] Huda Mahmoud,[g] Alain Van Dorsselaer,[a] Wael Ismail[e]

[a]Laboratoire de Spectrométrie de Masse Bio-Organique, Institut Pluridisciplinaire Hubert Curien, UMR7178 CNRS, Université de Strasbourg, Strasbourg, France
[b]Institut de Biologie Moléculaire des Plantes, CNRS, Université de Strasbourg, Strasbourg, France
[c]Département mécanique, ICube Laboratoire des Sciences de l'Ingénieur, de l'Informatique et de l'Imagerie, UNISTRA/CNRS/ENGEES/INSA, Strasbourg, France
[d]Institut für Mikrobiologie & Biotechnologie, Rheinische Friedrich-Wilhelms-Universität Bonn, Bonn, Germany
[e]Environmental Biotechnology Program, Life Sciences Department, College of Graduate Studies, Arabian Gulf University, Manama, Bahrain
[f]Faculty of Health and Life Sciences, Northumbria University, Newcastle upon Tyne, United Kingdom
[g]Department of Biological Sciences, Faculty of Science, Kuwait University, Kuwait City, Kuwait

**ABSTRACT** Sulfur metabolism in fuel-biodesulfurizing bacteria and the underlying physiological adaptations are not understood, which has impeded the development of a commercially viable bioprocess for fuel desulfurization. To fill these knowledge gaps, we performed comparative proteomics and untargeted metabolomics in cultures of the biodesulfurization reference strain *Rhodococcus qingshengii* IGTS8 grown on either inorganic sulfate or the diesel-borne organosulfur compound dibenzothiophene as a sole sulfur source. Dibenzothiophene significantly altered the biosynthesis of many sulfur metabolism proteins and metabolites in a growth phase-dependent manner, which enabled us to reconstruct the first experimental model for sulfur metabolism in a fuel-biodesulfurizing bacterium. All key pathways related to assimilatory sulfur metabolism were represented in the sulfur proteome, including uptake of the sulfur sources, sulfur acquisition, and assimilatory sulfate reduction, in addition to biosynthesis of key sulfur-containing metabolites such as S-adenosylmethionine, coenzyme A, biotin, thiamin, molybdenum cofactor, mycothiol, and ergothioneine (low-molecular weight thiols). Fifty-two proteins exhibited significantly different abundance during at least one growth phase. Sixteen proteins were uniquely detected and 47 proteins were significantly more abundant in the dibenzothiophene culture during at least one growth phase. The sulfate-free dibenzothiophene-containing culture reacted to sulfate starvation by restricting sulfur assimilation, enforcing sulfur-sparing, and maintaining redox homeostasis. Biodesulfurization triggered alternative pathways for sulfur assimilation different from those operating in the inorganic sulfate culture. Sulfur metabolism reprogramming and metabolic switches in the dibenzothiophene culture were manifested in limiting sulfite reduction and biosynthesis of cysteine, while boosting the production of methionine via the cobalamin-independent pathway, as well as the biosynthesis of the redox buffers mycothiol and ergothioneine. The omics data underscore the key role of sulfur metabolism in shaping the biodesulfurization phenotype and highlight potential targets for improving the biodesulfurization catalytic activity via metabolic engineering.

**IMPORTANCE** For many decades, research on biodesulfurization of fossil fuels was conducted amid a large gap in knowledge of sulfur metabolism and its regulation in fuel-biodesulfurizing bacteria, which has impeded the development of a commercially viable bioprocess. In addition, lack of understanding of biodesulfurization-associated metabolic and physiological adaptations prohibited the development of

Address correspondence to Wael Ismail, waelame@agu.edu.bh.

efficient biodesulfurizers. Our integrated omics-based findings reveal the assimilatory sulfur metabolism in the biodesulfurization reference strain *Rhodococcus qingshengii* IGTS8 and show how sulfur metabolism and oxidative stress response were remodeled and orchestrated to shape the biodesulfurization phenotype. Our findings not only explain the frequently encountered low catalytic activity of native fuel-biodesulfurizing bacteria but also uncover unprecedented potential targets in sulfur metabolism that could be exploited via metabolic engineering to boost the biodesulfurization catalytic activity, a prerequisite for commercial application.

**KEYWORDS** dibenzothiophene, sulfate starvation, cysteine biosynthesis, 4S pathway, mycothiol, sulfate activation complex

Several decades ago, microbial biodesulfurization emerged as a green process for removing sulfur from fossil fuels to accommodate environmental legislations, meet the ever-growing market demand for cleaner fuels, and overcome the technical and economic shortcomings of the conventional hydrodesulfurization process (1, 2). The concept was coined based on the unique metabolic capabilities of some bacteria that can utilize fuel-borne thiophenic organosulfur compounds as a sulfur source, thus reducing the total sulfur content of the biotreated fuel (1). The so-called "4S" pathway is the best-studied biodesulfurization mechanism (Fig. S1) (3, 4) that was originally elucidated in the actinobacterium *Rhodococcus qingshengii* IGTS8 (formerly *Rhodococcus erythropolis* IGTS8) (5–8). The 4S pathway selectively cleaves the carbon-sulfur bonds in the model diesel-borne organosulfur compound dibenzothiophene, eventually releasing the sulfur atom as sulfite (for assimilation) while preserving the carbon skeleton of dibenzothiophene as 2-hydroxybiphenyl (the end product of the pathway) (Fig. S1). The three key enzymes of the 4S pathway (DszC, DszA, and DszB) are encoded in the *dsz* operon on a 120-kb linear plasmid in the IGTS8 strain (9, 10). DszC is a monooxygenase that initiates the pathway by transforming dibenzothiophene to dibenzothiophene sulfoxide and subsequently to dibenzothiophene sulfone. The latter is the substrate of the second monooxygenase DszA, which cleaves one of the C-S bonds of the thiophene ring to produce 2-hydroxybiphenyl-2′-sulfinate. The last and sulfur-releasing reaction is catalyzed by DszB, a desulfinase that produces sulfite and 2-hydroxybiphenyl. A chromosomally encoded flavin reductase supplies the two monooxygenases with $FMNH_2$ (1, 2). Despite intensive research on the system, a commercially viable biodesulfurization technology for the oil industry could not be established yet, mainly due to the very low catalytic activity and insufficient robustness of the applied biocatalysts/microbial hosts, among other hurdles (1, 2). Moreover, it is currently unknown how naturally occurring biodesulfurizing bacteria import dibenzothiophene and excrete 2-hydroxybiphenyl. Some studies reported dibenzothiophene uptake in recombinant strains by an ABC-type transporter (11–13).

So far, biodesulfurization research has focused on characterizing the 4S pathway and improving its biocatalytic efficiency, and indeed a great deal of knowledge has accumulated (2, 3, 14). However, much less attention has been paid to the direct connection of the biodesulfurization process with the assimilatory sulfur metabolism, which is tightly interwoven with other indispensable metabolic and physiological processes (15). This has left huge gaps in our understanding of assimilatory sulfur metabolism in biodesulfurizing microbes, which is, however, mandatory when aiming at the development of a commercially viable biodesulfurization technology. When challenged with less preferred substrates, such as dibenzothiophene, as the sole sulfur source, the biodesulfurizing bacteria will face sulfate starvation conditions (16). It is currently unknown how biodesulfurizing bacteria respond to this stressor and how they might remodel their assimilatory sulfur metabolism under biodesulfurization conditions (1, 17–19).

Evidence is accumulating that some factors in biodesulfurizing bacteria, apart from the 4S pathway, are key determinants of the biodesulfurization efficiency, including

some enzymes of sulfur metabolism such as sulfite reductase and cystathionine-$\beta$-synthase (1, 17, 20). Accordingly, studying the biodesulfurization phenotype beyond the 4S pathway and thorough understanding of sulfur metabolism and its regulation in biodesulfurizing microbes is key to elucidate how they respond/adapt to the sulfate-limiting conditions that prevail when diesel-borne organosulfur compounds, like dibenzothiophene, are provided as the sole sulfur source (17, 18). While systems biology approaches have been applied successfully to study sulfur metabolism in *Escherichia coli* (21, 22), *Pseudomonas* spp. (23, 24), and *Bacillus subtilis* (25, 26), such studies are rare for fuel-biodesulfurizing bacteria and provided insights only into details of dibenzothiophene and benzothiophene desulfurization (27, 28).

To fill these knowledge gaps, we performed proteomics and metabolomics studies with *R. qingshengii* IGTS8 and compared cultures grown on either dibenzothiophene or inorganic sulfate. The data enabled us to build an experimentally supported model for sulfur metabolism in *R. qingshengii* IGTS8 and unveil biodesulfurization-driven adaptive responses. We show how the biodesulfurization-associated sulfate starvation cues provoke sulfur metabolism remodeling and suggest systems for dibenzothiophene uptake and efflux of the biodesulfurization product 2-hydroxybiphenyl. In addition, we identify a probable global regulator of sulfur metabolism in the IGTS8 strain. Our omics data reveal metabolic engineering hot spots in sulfur metabolism that could be manipulated to design novel recombinant strains having enhanced biodesulfurization activity.

## RESULTS

**A global look at the sulfur proteome and metabolome.** A total of 2,896 out of the 6,734 proteins encoded by the IGTS8 genome were confidently identified by proteomic analyses. Among those, we identified several sulfur metabolism proteins (the sulfur proteome) with significant changes in their abundance depending on the sulfur source and growth phase, while the level of other sulfur metabolism proteins did not vary significantly (Table S1). In addition, a few sulfur metabolism proteins were detected but could not be quantified, and a smaller number could not be detected in either of the cultures. In total, 29 metabolites related to sulfur metabolism (sulfur metabolome) were detected (Table S3). These are mainly metabolites of dibenzothiophene desulfurization (4S pathway) and biosynthesis of cysteine, methionine, coenzyme A (CoA), biotin, mycothiol, ergothioneine, thiamin, and molybdenum cofactor (MoCo), in addition to S-adenosylmethionine metabolism and sulfur relay pathways. The highest change in abundance was observed for metabolites of dibenzothiophene desulfurization, mycothiol biosynthesis, and thiamin and S-adenosylmethionine metabolism. The metabolome coverage was 16 to 19% depending on the growth phase (see the supplemental material for proteins and metabolites numbers).

Principal-component analysis (PCA) (Fig. 1) showed that the 16 samples from the dibenzothiophene culture (representing four growth phases) are clustered distinctly from the corresponding samples of the inorganic sulfate culture, attesting for the uniqueness of the sulfur proteome and metabolome of both cultures. Moreover, in the biodesulfurizing culture, there was a clear grouping of samples from each growth phase in fairly separated clusters, an indication of temporal adaptation, which was not the case in the inorganic sulfate culture. Based on the proteomics and metabolomics data, we proposed a model for sulfur metabolism in the IGTS8 strain as presented in the following parts of the results (Fig. 2). The model depicts sulfur assimilation pathways under both sulfate-rich and sulfate-deficient (biodesulfurization) conditions and covers sulfur source uptake, sulfur acquisition, sulfate/sulfite reduction, biosynthesis of cysteine, methionine, and S-adenosylmethionine, and sulfuryl group transfer for the biosynthesis of sulfated metabolites.

**Uptake of the sulfur source.** The first and most probable candidate for sulfate import is the SulT superfamily member CysPTWA/Sbp transporter (IGTS8_peg2391 to IGTS8_peg2395) (Fig. 2 and 3, Table 1, and Table S1). Components of this transporter were identified in both the dibenzothiophene and sulfate cultures. The ATP-binding protein (CysA) was 8.7-fold more abundant ($\log_2$ fold change = 3.1) in the dibenzothiophene

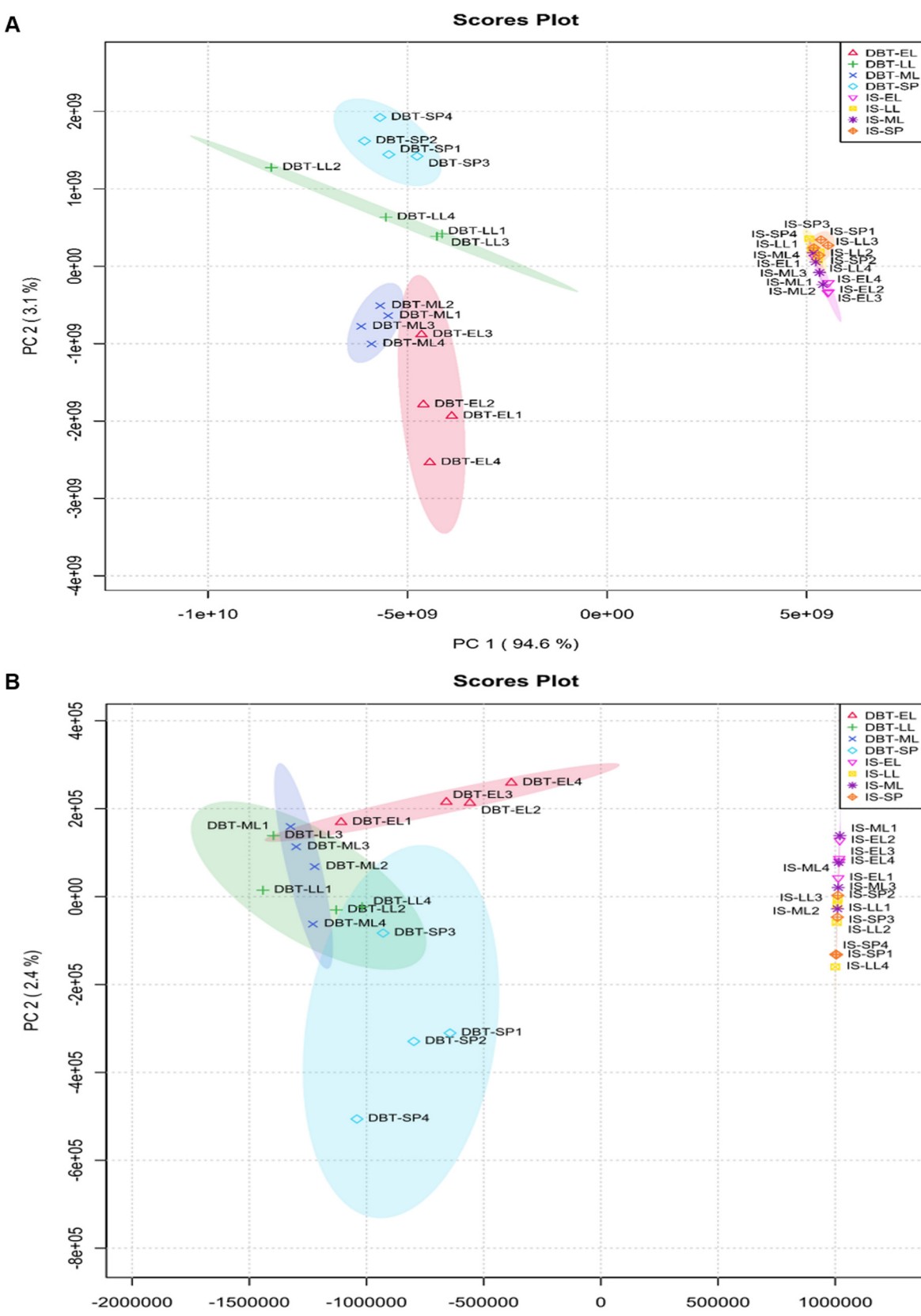

**FIG 1** Principal-component analysis for (A) proteins and (B) metabolites related to sulfur metabolism using the two main dimensions. DBT indicates the dibenzothiophene culture and IS indicates the inorganic sulfate culture. The growth phases are abbreviated as EL (early log), ML (mid-log), LL (late log), and SP (stationary phase). The number shown after the growth phase indicates the number of the replicates.

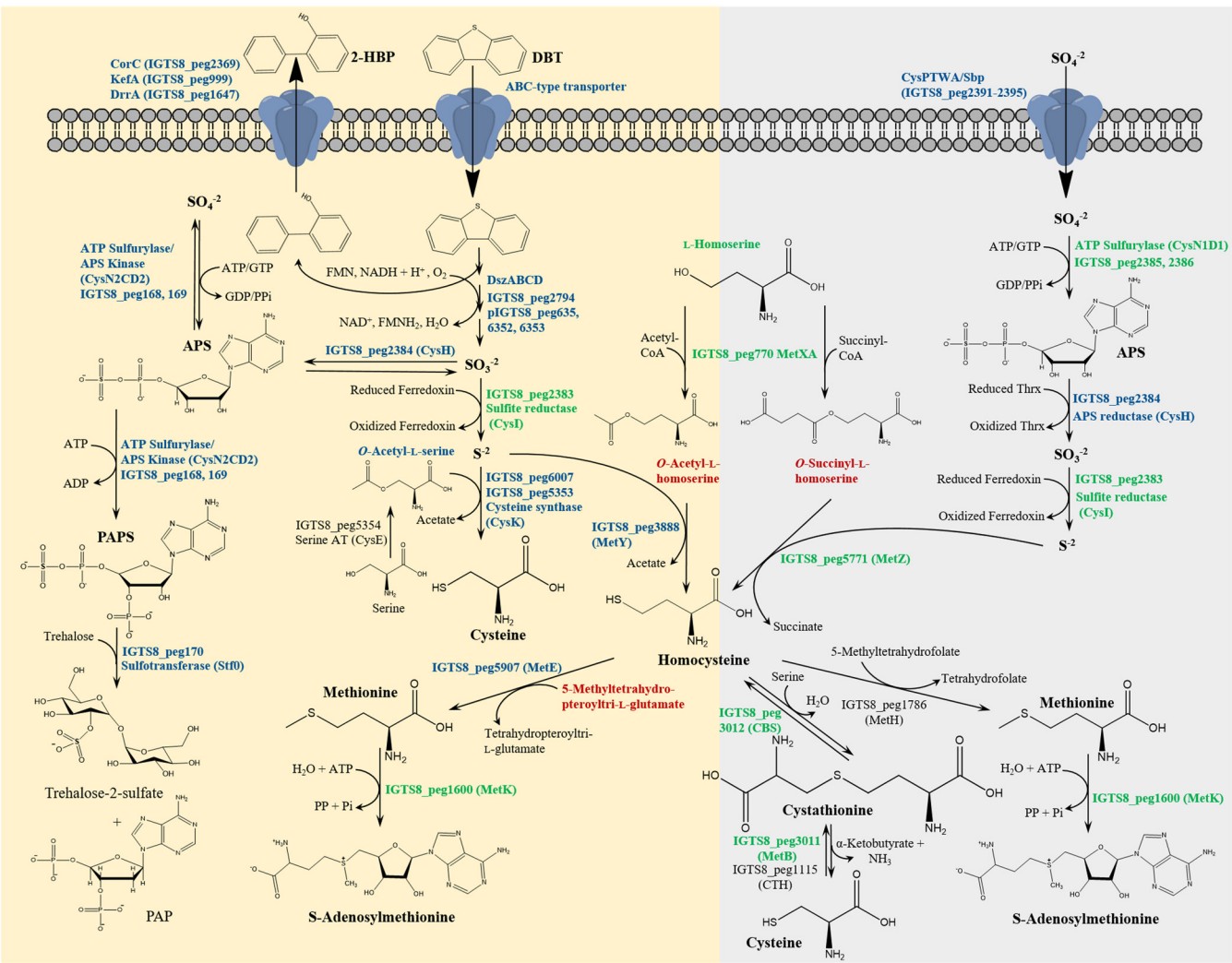

**FIG 2** A proposed model for assimilatory sulfur metabolism in *R. qingshengii* IGTS8 under both sulfate starvation (biodesulfurization, highlighted in light yellow) and sulfate-rich (highlighted in light gray) conditions. Proteins and metabolites in blue font are significantly more abundant in the dibenzothiophene culture at least during one growth phase, while those appearing in red font are significantly more abundant in the inorganic sulfate culture at least during one growth phase. The abundance of proteins and metabolites appearing in green font was not significantly different between the dibenzothiophene and inorganic sulfate cultures. Proteins in black font were either not detected or detected but could not be quantified (see Table 1 and Tables S1 and S3 for details of the abundance profiles). PAP, 3′-phosphoadenosine 5′-phosphate.

culture during the mid-log phase. In contrast, the level of the substrate-binding component CysP/SbP was not significantly different between both cultures. In this transporter, the *cysT* gene occurs twice (IGTS8_peg2393 and IGTS8_peg2394) (Fig. 3).

The proteome of the biodesulfurizing culture was rich in proteins involved in uptake (ABC-type transporters) and utilization (sulfatases and oxygenases) of low-preference sulfur sources such as sulfate esters and sulfonates during all growth phases (Table 1 and Table S1). Genes encoding the detected ABC-type transporters appear to be coexpressed from putative operons, and the proteins were either highly enriched (up to 257-fold, $\log_2$ fold change = 2.5 to 7) or uniquely present in the dibenzothiophene culture (Fig. 3). It is, therefore, likely that one or more of those transport systems could play a role in dibenzothiophene import. Uptake of dibenzothiophene and its alkylated derivatives by an ABC-type transporter was shown in recombinant strains (11–13) but has not been reported to date in native biodesulfurizing bacteria.

**Sulfate availability stimulates divergent routes for sulfate/sulfite reduction. Sulfite is a metabolic branching point.** In the sulfate-grown culture, sulfate was reduced via the classical assimilatory sulfate reduction route, in line with the detection of

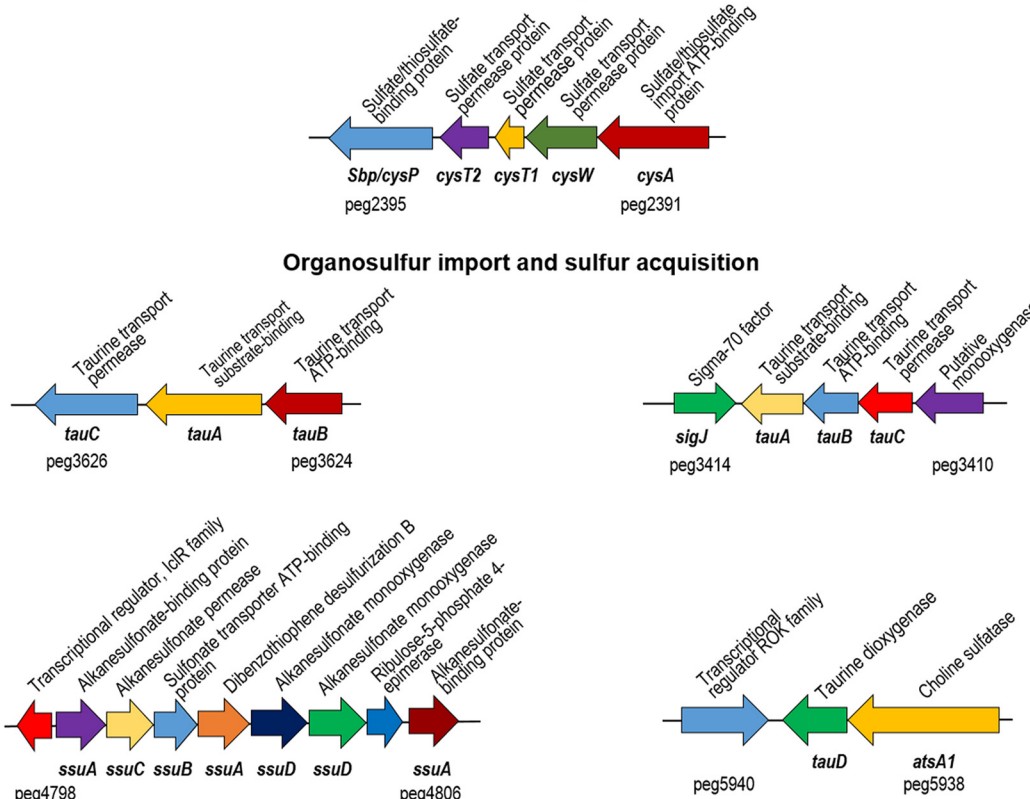

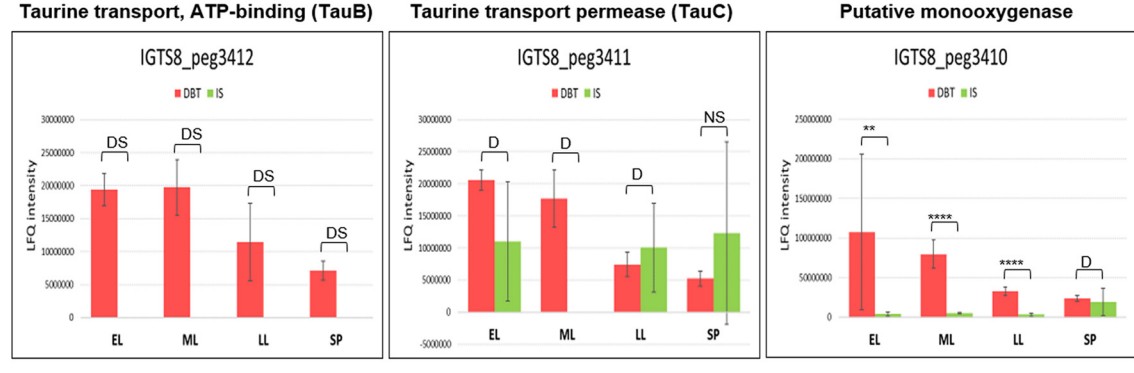

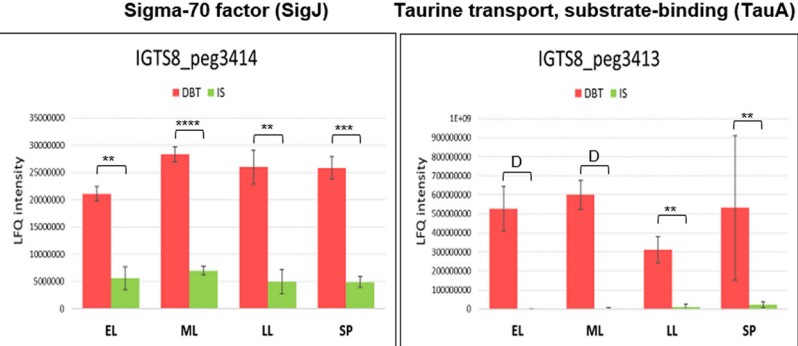

**FIG 3** Gene clusters and proteins of sulfur source uptake and sulfur acquisition (see Table S1 for details of the abundance profiles and proposed functions of the proteins). Protein annotations are shown above the gene clusters, and gene names and IDs of the

the enzymes of assimilatory sulfate reduction in the proteome, namely, ATP sulfurylase (sulfate adenylyltransferase, CysN1D1), adenylylsulfate (APS) reductase (CysH), and a one-subunit cyanobacterial-type ferredoxin-sulfite reductase (CysI) (Fig. 2, Table 1, and Table S1). Downstream of the sulfate adenylyltransferase gene and in the same operon, we found a gene (IGTS8_peg2387) likely encoding SirB that catalyzes the ferro-chelation of sirohydrochlorin to siroheme, the prosthetic group of sulfite reductase. The five enzymes are encoded in an operon (Fig. 4), a gene organization that is different from that of *Escherichia coli* and *Bacillus subtilis* where the assimilatory sulfate reduction genes are located in several transcriptional units (29). The gene cluster is transcribed in the opposite direction to the nearby *cysPT$_1$T$_2$WA/sbp* cluster encoding the sulfate uptake system, an organization reminiscent of that of *Mycobacterium* spp. (30). While SirB was not detected, the abundance of CysD1N1 and CysI was not significantly different between the dibenzothiophene and sulfate cultures. The level of only CysH appeared to increase slightly in the dibenzothiophene culture toward the stationary phase (log$_2$ fold change = 2.1) (Table S1). Signature sequences of APS reductases (CCALRKVAPL, SIGCAPCTS, KTECGLH) that are lacking in phosphoadenosine phosphosulfate (PAPS) reductase activity are present in the CysH protein sequence (31, 32).

In the dibenzothiophene culture, sulfur acquisition starts with the release of the sulfur atom from dibenzothiophene as sulfite via the 4S pathway (Fig. 2). Indeed, enzymes of the 4S pathway (DszABCD) were much more abundant (up to 380-fold) in the dibenzothiophene culture during all growth phases, and the relative abundance of the four Dsz proteins to each other remained almost constant during all growth phases (Fig. 5, Table 1, and Table S1). We show here for the first time the temporal shift of the Dsz enzymes' abundance evaluated by nanoscale liquid chromatography-tandem mass spectrometry (nanoLC-MS/MS) analysis. Consistent with the identification of the Dsz enzymes, the biodesulfurizing culture had up to 519-fold (log$_2$ fold change = 6.7 to 9.0) higher content of the 4S pathway intermediates dibenzothiophene sulfoxide and dibenzothiophene sulfone which were uniquely present in the dibenzothiophene-grown cells during the early log, mid-log, and stationary phases (Fig. 5, Table 2, and Table S3).

Sulfite released from dibenzothiophene could be reduced to sulfide directly with ferredoxin-sulfite reductase (CysI) without prior activation. Alternatively, sulfite could be oxidized by sulfite oxidases and oxidoreductases to sulfate, excreted, and reimported for reduction and assimilation as assumed by Aggarwal et al. (17). In *R. qingshengii* IGTS8, one candidate gene encoding sulfite oxidase/oxidoreductase was identified (IGTS8_peg2618). It is a membrane-bound molybdoenzyme (five transmembrane helices) bearing resemblance to the well-characterized SoxC subunit of sulfane dehydrogenase from *Paracoccus pantotrophus* (33). The most similar structurally characterized proteins are the SorA subunit of *Starkeya novella* sulfite dehydrogenase (34), followed by *P. pantotrophus* SoxC and SorT sulfite dehydrogenase from *Sinorhizobium meliloti* (35). Moreover, IGTS8_peg446 encodes a putative membrane protein YeiH with 11 transmembrane helices that is 28% identical to a YeiH family sulfate exporter (TDL75784) from *R. qingshengii* S-E5. However, these two proteins were not detected in the IGTS8 proteome. The exclusive presence of subunit 2 of a second ATP sulfurylase (CysD2, IGTS8_peg169) during all growth phases of the dibenzothiophene culture was unexpected and suggested an additional route for sulfite lacking in the sulfate culture (Fig. 2 and 4), highlighting sulfite as a metabolic branching point.

Subunit 1 (CysN2) of this ATP sulfurylase (IGTS8_peg168) is fused to adenylylsulfate kinase (CysC), encoded upstream of *cysD2*, and was also detected but could not be properly quantified. Together, CysN2CD2 proteins likely constitute a bifunctional sul-

**FIG 3** Legend (Continued)

first and last genes are shown below the gene clusters. The growth phases are abbreviated as EL (early log), ML (mid-log), LL (late log), and SP (stationary phase). Bar charts represent the label-free quantification (LFQ) values showing the abundance profile of proteins encoded by a putative taurine transport operon in both the DBT (dibenzothiophene) and IS (inorganic sulfate) cultures. Significance of the data is attested by a Welsch moderated *t* test as follows: NS for $P > 0.05$, * for $P \leq 0.05$, ** for $P \leq 0.01$, *** for $P \leq 0.001$, **** for $P \leq 0.0001$, DS for a protein which was uniquely identified in the dibenzothiophene cultures but not detected in the sulfate cultures, D for a protein which was identified but not confidently quantified.

**TABLE 1** Proteins[a] of sulfur metabolism showing significantly different abundance between the dibenzothiophene and sulfate cultures

| Protein ID | Annotation | Protein name | Proposed function/pathway |
|---|---|---|---|
| **Proteins upregulated in the dibenzothiophene culture** | | | |
| IGTS8_peg167 | Probable sulfatase | | Sulfur acquisition |
| IGTS8_peg169 | Sulfate adenylyltransferase subunit 2 | CysD | Assimilatory sulfate reduction/sulfate activation complex |
| IGTS8_peg170 | Sulfotransferase | Stf0 | Sulfuryl group transfer/biosynthesis of sulfated metabolites |
| IGTS8_peg394 | Predicted transcriptional regulator of sulfate adenylyltransferase | CymR/Rrf2 | Probable transcriptional regulator for cysteine metabolism |
| IGTS8_peg999 | Potassium efflux system KefA protein/small-conductance mechanosensitive channel | KefA | Probable 2-HBP efflux |
| IGTS8_peg1417 | Alkylhydroperoxide reductase protein C | AhpC | Oxidative stress response |
| IGTS8_peg1647 | ABC-type multidrug transport system, ATPase component | DrrA_5 | Probable 2-HBP efflux |
| IGTS8_peg2369 | Magnesium and cobalt efflux protein CorC | CorC | Probable 2-HBP efflux |
| IGTS8_peg2391 | Sulfate and thiosulfate import ATP-binding protein | CysA | Sulfate uptake |
| IGTS8_peg2384 | Phosphoadenylylsulfate reductase (thioredoxin)/adenylylsulfate reductase (thioredoxin) | CysH | Assimilatory sulfate reduction |
| IGTS8_peg2794 | Nitrilotriacetate monooxygenase component B | DszD | Flavin reductase (4S pathway) |
| IGTS8_peg2690 | Alpha-ketoglutarate-dependent taurine dioxygenase | TauD_7 | Sulfur acquisition |
| IGTS8_peg2683 | Luciferase family protein | SsuD | Sulfur acquisition |
| IGTS8_peg2998 | Pantothenate kinase | CoaA | CoA biosynthesis |
| IGTS8_peg3074 | Dimethylhistidine N-methyltransferase | EgtD | Ergothioneine biosynthesis |
| IGTS8_peg3075 | Glutamine amidotransferases class-II | EgtC | Ergothioneine biosynthesis |
| IGTS8_peg3077 | Glutamate-cysteine ligase | EgtA | Ergothioneine biosynthesis |
| IGTS8_peg3153 | Molybdenum cofactor biosynthesis protein | MoaB | Molybdenum cofactor biosynthesis |
| IGTS8_peg3410 | Putative monooxygenase | | Sulfur acquisition from sulfonates |
| IGTS8_peg3412 | Taurine transport ATP-binding protein | TauB | ABC transporter |
| IGTS8_peg3413 | Possible ABC sulfonate transporter, substrate–binding component | TauA | ABC transporter |
| IGTS8_peg3414 | RNA polymerase sigma–70 factor | SigJ | Transcriptional regulator |
| IGTS8_peg3535 | Nitrilotriacetate monooxygenase component A | | Probable sulfur acquisition protein |
| IGTS8_peg3624 | ABC-type nitrate/sulfonate/bicarbonate transport system, ATPase component | TauB | ABC transporter |
| IGTS8_peg3625 | Taurine-binding periplasmic protein | TauA | ABC transporter |
| IGTS8_peg3800 | Methionine ABC transporter substrate-binding protein | MetQ | Methionine transport |
| IGTS8_peg3801 | Methionine ABC transporter ATP-binding protein | MetN | Methionine transport |
| IGTS8_peg3888 | O-Acetylhomoserine sulfhydrylase/O-succinylhomoserine sulfhydrylase | MetY | Cysteine and methionine metabolism |
| | Homoserine O-acetyltransferase | | |
| IGTS8_peg4062 | Alkanesulfonate monooxygenase | SsuD | Sulfur acquisition |
| IGTS8_peg4243 | Coenzyme F420-dependent N5,N10-methylene tetrahydromethanopterin reductase and related flavin-dependent oxidoreductases; sulfonate monooxygenase | SfnG | Sulfur acquisition |
| IGTS8_peg4799 | Alkanesulfonates-binding protein | SsuA | Alkanesulfonate transporter |
| IGTS8_peg4801 | Alkanesulfonates ABC transporter ATP-binding protein | SsuB | Alkanesulfonate transporter |
| IGTS8_peg4802 | Dibenzothiophene desulfurization enzyme B | SsuA/DszB | Sulfur acquisition |
| IGTS8_peg4804 | Alkanesulfonate monooxygenase | SsuD | Sulfur acquisition |
| IGTS8_peg4806 | Alkanesulfonates-binding protein | SsuA | Alkanesulfonate transporter |
| IGTS8_peg4968 | Ketopantoate reductase PanG | PanG | CoA biosynthesis |

**TABLE 1** (Continued)

| Protein ID | Annotation | Protein name | Proposed function/pathway |
|---|---|---|---|
| IGTS8_peg5732 | Nitrilotriacetate monooxygenase component A | ScmK | Probable sulfur acquisition/cysteine biosynthesis protein |
| IGTS8_peg5783 | Methionine ABC transporter ATP-binding protein | TcyC | Probable cystine transporter |
| IGTS8_peg5784 | Periplasmic binding protein | FliY/TcyA | Probable cystine transporter |
| IGTS8_peg5785 | Sarcosine oxidase | SoxA_1 | Glycine, serine, threonine metabolism |
| IGTS8_peg5907 | 5-Methyltetrahydropteroyltriglutamate-homocysteine methyltransferase (cobalamin-independent) | MetE | Methionine biosynthesis |
| IGTS8_peg5938 | Choline sulfatase | AtsA1 | Sulfur acquisition |
| IGTS8_peg5939 | Alpha-ketoglutarate-dependent taurine dioxygenase | TauD | Sulfur acquisition |
| IGTS8_peg5940 | Possible transcriptional regulator, ROK family | | Transcriptional regulator |
| IGTS8_peg5948 | Glycosyltransferase MshA involved in mycothiol biosynthesis | MshA | Mycothiol biosynthesis |
| IGTS8-peg6007 | Cysteine synthase B | CysK | Cysteine biosynthesis |
| pIGTS8_peg6351 | Dibenzothiophene desulfurization enzyme A | DszA | Sulfur acquisition from DBT (4S pathway) |
| pIGTS8_peg6352 | Dibenzothiophene desulfurization enzyme B | DszB | Sulfur acquisition from DBT (4S pathway) |
| pIGTS8_peg6353 | Acyl-CoA dehydrogenase; probable dibenzothiophene desulfurization enzyme | DszC | Sulfur acquisition from DBT (4S pathway) |
| **Proteins downregulated in the dibenzothiophene culture** | | | |
| IGTS8_peg751 | Possible rhodanese-related sulfurtransferase | PspE | Sulfurtransferase |
| IGTS8_peg1918 | Thiosulfate sulfurtransferase, rhodanese | TrxA | Thioredoxin |
| IGTS8_peg4196 | Putative hydrolase | | Probable mycothiol metabolism protein |
| IGTS8_peg4995 | 2-Hydroxychromene-2-carboxylate isomerase/DsbA-like thioredoxin domain | DsbA | Protein-disulfide isomerase |

aProteins having log2 fold change greater than 2.0 (upregulated) or less than −2.0 (downregulated) and a P value of <0.05. See Table S1 for more information on protein abundance profiles and protein families.

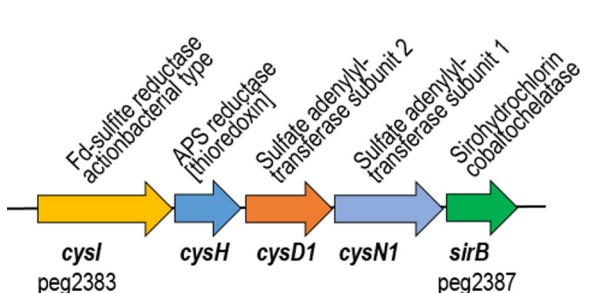

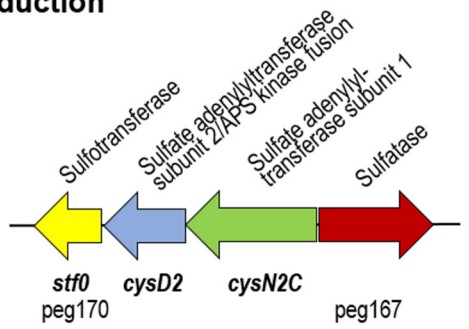

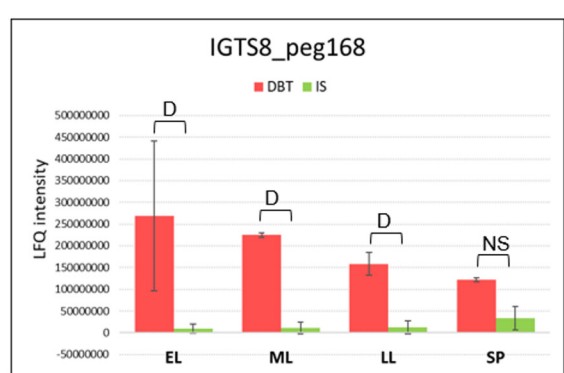

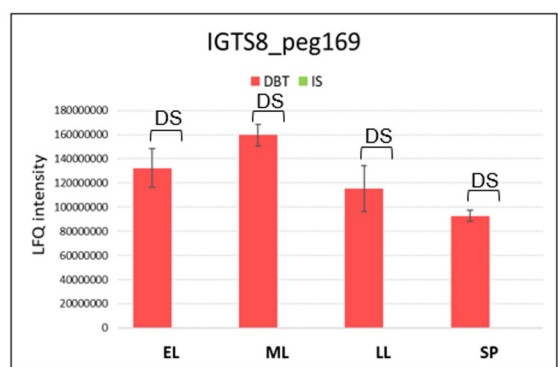

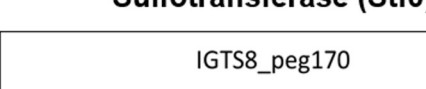

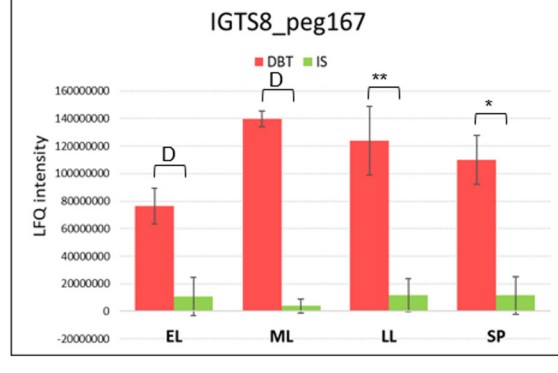

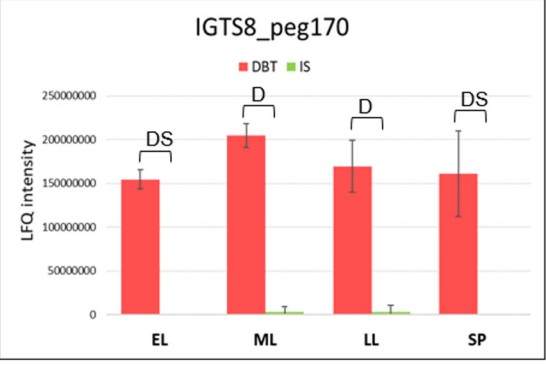

**FIG 4** Gene clusters and proteins of sulfate/sulfite activation and reduction (see Table S1 for details of the abundance profiles and proposed functions of the proteins). Protein annotations are shown above the gene clusters, and gene names and IDs of the first and last genes are shown below the gene clusters. The growth phases are abbreviated as EL (early log), ML (mid-log), LL (late log), and SP (stationary phase). Bar charts represent the label-free quantification (LFQ) values showing the abundance profile of the proteins in both the DBT (dibenzothiophene) and IS (inorganic sulfate) cultures. Significance of the data is attested by a Welch moderated $t$ test as follows: NS for $P > 0.05$, * for $P \leq 0.05$, ** for $P \leq 0.01$, *** for $P \leq 0.001$, **** for $P \leq 0.0001$, DS for a protein which was uniquely identified in the dibenzothiophene cultures but not detected in the sulfate cultures, D for a protein which was identified but not confidently quantified.

fate adenylyltransferase/APS kinase sulfate activation complex that may catalyze the activation of sulfate to APS and then to PAPS (Fig. 2). A sulfotransferase (Stf0, IGTS8_peg170) that utilizes PAPS as the sulfuryl group donor for the biosynthesis of sulfated metabolites is encoded downstream of *cysD2*. Notably, this enzyme was found exclusively in the early log and stationary phase cultures of dibenzothiophene (Fig. 2 and 4, Table 1, and Table S1).

**Biodesulfurization restricted cysteine production and boosted methionine biosynthesis.** The *R. qingshengii* IGTS8 genome encodes various pathways and enzyme paralogs for cysteine, homocysteine, and methionine biosynthesis (29, 36, 37). Cysteine biosynthesis enzymes of the direct sulfhydrylation and reverse transsulfuration

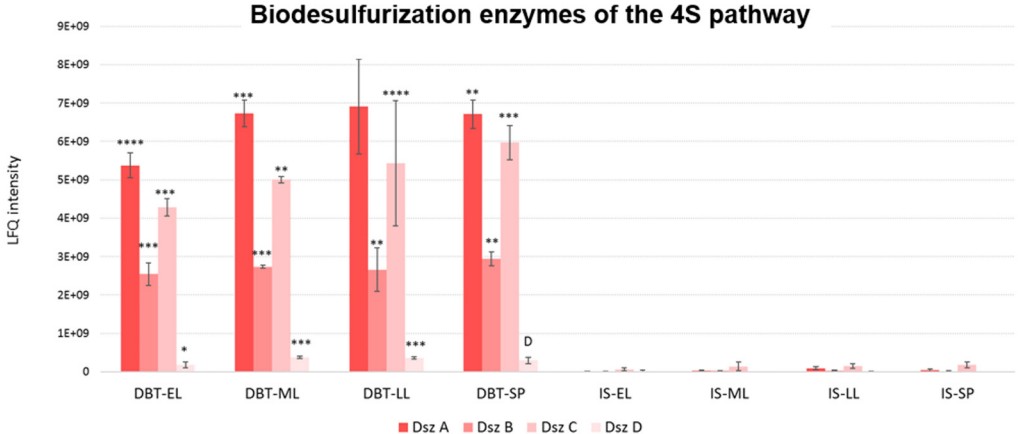

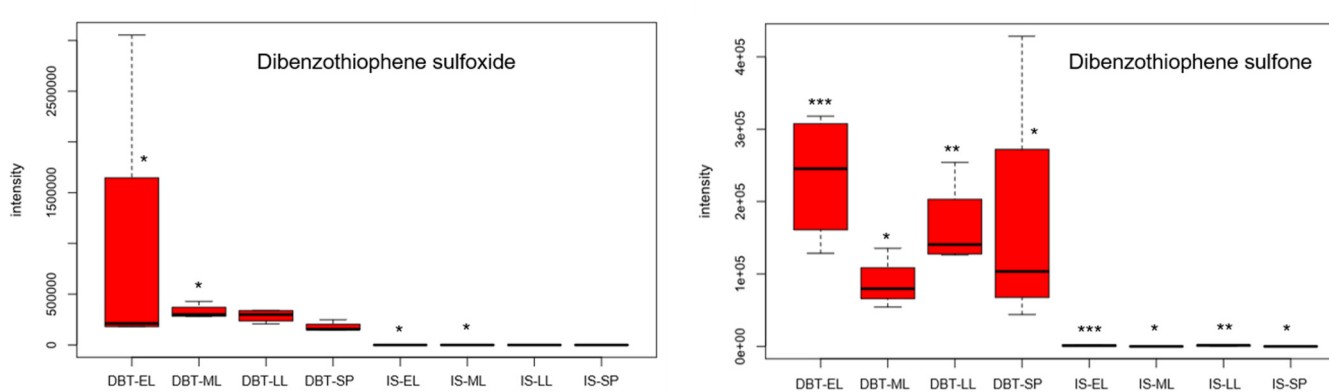

**FIG 5** Proteins and metabolites of dibenzothiophene biodesulfurization via the 4S pathway (see Table S1 for details of the abundance profiles and proposed functions of the proteins). The growth phases are abbreviated as EL (early log), ML (mid-log), LL (late log), and SP (stationary phase). Bar charts represent the label-free quantification (LFQ) values showing the abundance profile of the proteins in both the DBT (dibenzothiophene) and IS (inorganic sulfate) cultures. Significance of the data is attested by a Welch moderated $t$ test as follows: NS for $P > 0.05$, * for $P \leq 0.05$, ** for $P \leq 0.01$, *** for $P \leq 0.001$, **** for $P \leq 0.0001$, DS for a protein which was uniquely identified in the dibenzothiophene cultures but not detected in the sulfate cultures, D for a protein which was identified but not confidently quantified. Metabolomics data are shown as boxplots displaying the distribution for each growth phase with the minimum, maximum, and median values for the dibenzothiophene (DBT) and inorganic sulfate (IS) cultures. Significance of the data ($P$ value [rank], Wilcoxon test) is indicated by asterisks: * for $P < 0.01$, ** for $P < 0.05$, *** for $P < 0.1$, no asterisk for $P > 0.1$.

pathways (29) were identified in the proteome but were not among the highly abundant or *de novo* induced proteins in the dibenzothiophene culture (Table S1). Still, the relative abundance of the proteins together with the metabolomics data underscore the possibility that cysteine biosynthesis in the dibenzothiophene and sulfate cultures might proceed via distinct (more preferred) pathways (Fig. 2, Tables 1 and 2, and Table S1). The level of the cysteine synthase CysK (IGTS8_peg6007) was slightly higher (log$_2$ fold change = 1.0) in the dibenzothiophene culture and was significantly more abundant during the stationary phase (log$_2$ fold change = 2.1), suggesting that direct sulfhydrylation might be the preferred route of cysteine biosynthesis under biodesulfurization conditions. The detection of a higher content of *O*-acetyl-L-serine (log$_2$ fold change = 0.27 to 1.35), the substrate of CysK, in the dibenzothiophene culture substantiates this assumption (Fig. 6 and Table S3). A serine acetyltransferase-encoding gene (CysE, IGTS8_peg5353) was identified in the IGTS8 genome downstream of the gene for a second cysteine synthase CysK1 (IGTS8_peg5354). The gene product was detected in the proteome (Table S1) but could not be quantified.

In the sulfate culture, the reverse transsulfuration pathway was probably the primary route for cysteine biosynthesis (Fig. 2), which is indicated by slightly higher, though not significantly different, levels of the pyridoxal phosphate-dependent enzymes cystathionine-$\beta$-synthase (CBS, IGTS8_peg3012) and cystathionine-$\gamma$-lyase

**TABLE 2** Metabolites[a] of sulfur metabolism showing significantly different abundance between the dibenzothiophene and sulfate cultures

| Metabolite name | Proposed function/pathway |
| --- | --- |
| Metabolites that were more abundant in the dibenzothiophene culture | |
| Dibenzothiophene sulfoxide | Dibenzothiophene desulfurization, 4S pathway |
| Dibenzothiophene sulfone | Dibenzothiophene desulfurization, 4S pathway |
| O-Phosphohomoserine | Cysteine and methionine biosynthesis |
| O-Acetyl-L-serine | Cysteine biosynthesis |
| L-Histidinol | Ergothioneine biosynthesis |
| 1-O-(2-Amino-1-deoxy-α-D-glucopyranosyl)-D-myo-inositol | Mycothiol biosynthesis |
| 2-Iminoacetate | Thiamin/thiazole metabolism |
| 1-Aminocyclopropane-1-carboxylic acid | S-Adenosylmethionine metabolism |
| 5'-Deoxyadenosine | Cleavage product of S-adenosylmethionine |
| Coenzyme A | CoA and pantothenate biosynthesis |
| | |
| Metabolites that were more abundant in the inorganic sulfate culture | |
| O-Succinyl-L-homoserine | Cysteine and methionine biosynthesis |
| O-Acetyl-L-homoserine | Cysteine biosynthesis |
| Biotin | Biotin biosynthesis |
| Mycothiol | Mycothiol biosynthesis |
| Dephospho-CoA | CoA and pantothenate biosynthesis |
| Cyclic pyranopterin monophosphate | Folate/molybdenum cofactor biosynthesis/sulfur relay pathways, known as precursor Z |

[a]Metabolites having $\log_2$ fold change greater than 1.0 or less than −1.0. See Table S3 for more information on the metabolite abundance profile.

(MetB, IGTS8_peg3011) in the sulfate culture (Table S1). In addition, the sulfate culture had a significantly higher content of the homocysteine precursors O-acetyl-L-homoserine ($\log_2$ fold change = −1.16 to −1.96) and O-succinyl-L-homoserine ($\log_2$ fold change = −1.82 to −2.61) (Fig. 6 and Table S3), pointing to a potentially higher homocysteine biosynthetic activity to feed into the reverse transsulfuration pathway. Other possibilities for cysteine biosynthesis in *R. qingshengii* IGTS8 using phosphoserine or phosphohomoserine may be envisaged (see supplemental material for details).

Unlike cysteine, biodesulfurization probably triggered more methionine biosynthesis as indicated by the increased production of MetE in the dibenzothiophene culture (Fig. 2, Tables 1 and 2, and Tables S1 and S3). Moreover, homocysteine (methionine precursor) was apparently produced in both the dibenzothiophene and sulfate cultures via divergent routes (Fig. 2). A homoserine acyltransferase (MetXA) was detected in the proteome of both cultures, and therefore it could similarly catalyze acylation of homoserine. Although the products of the MetXA-catalyzed reactions, O-acetyl-L-homoserine and O-succinyl-L-homoserine, were both more abundant in the sulfate culture, the dibenzothiophene culture appeared to preferentially utilize O-acetyl-L-homoserine, which is consistent with the significantly higher abundance of O-acetylhomoserine sulfhydrylase (MetY, IGTS8_peg3888) under biodesulfurization conditions ($\log_2$ fold change = 2.9 to 3.1). In the sulfate culture, a MetY isoenzyme (MetZ), whose abundance did not vary significantly, potentially transformed O-succinyl-L-homoserine to homocysteine (Fig. 2 and 6, Tables 1 and 2, and Tables S1 and S3).

Under biodesulfurization conditions, methionine was most probably produced from homocysteine via MetE, the cobalamin-independent 5-methyltetrahydropteroyltri-L-glutamate-homocysteine methyltransferase. This protein is one of the top-10 most abundant proteins in the sulfur proteome of the biodesulfurizing culture, and its level was maximum during the early log phase (Fig. 6, Table 1, and Table S1). Compared to that in the inorganic sulfate culture, the level of this protein was 190- to 273-fold higher in the biodesulfurizing culture depending on the growth phase. Consistent with this finding, the methyl group donor for MetE (5-methyltetrahydropteroyltri-L-glutamate) was more abundant in the dibenzothiophene culture during the early log phase ($\log_2$ fold change = 1.9), and its level steadily decreased afterwards to become more abundant in the sulfate culture during the late log and stationary phases ($\log_2$ fold change = −2.66 and −4.07, respectively) (Fig. 6 and Table S3). To the contrary, in the presence of the preferred sulfur

**A**

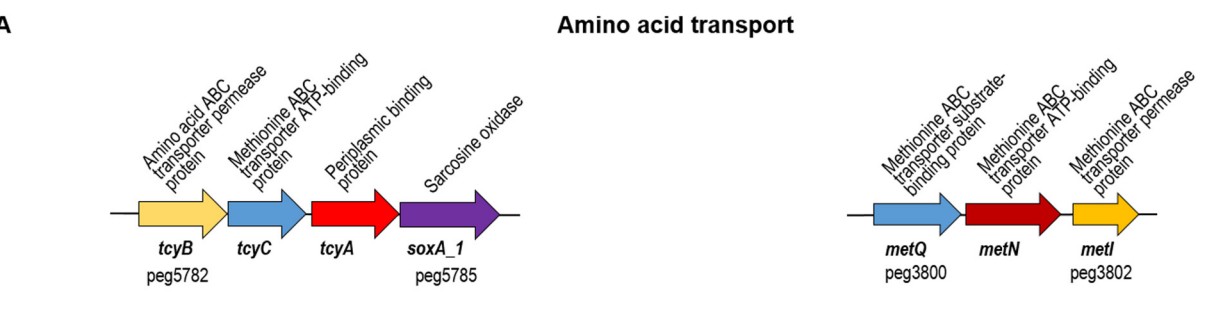

**Amino acid transport**

**B**

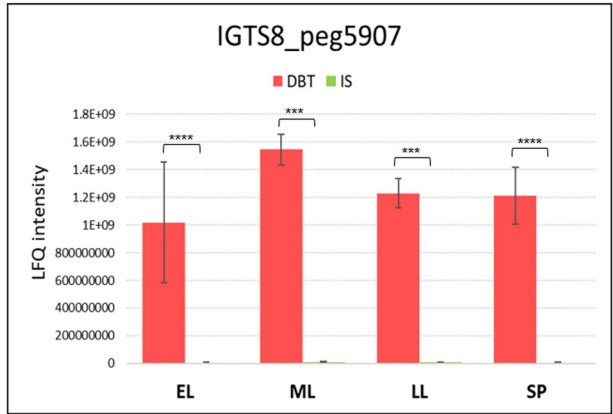

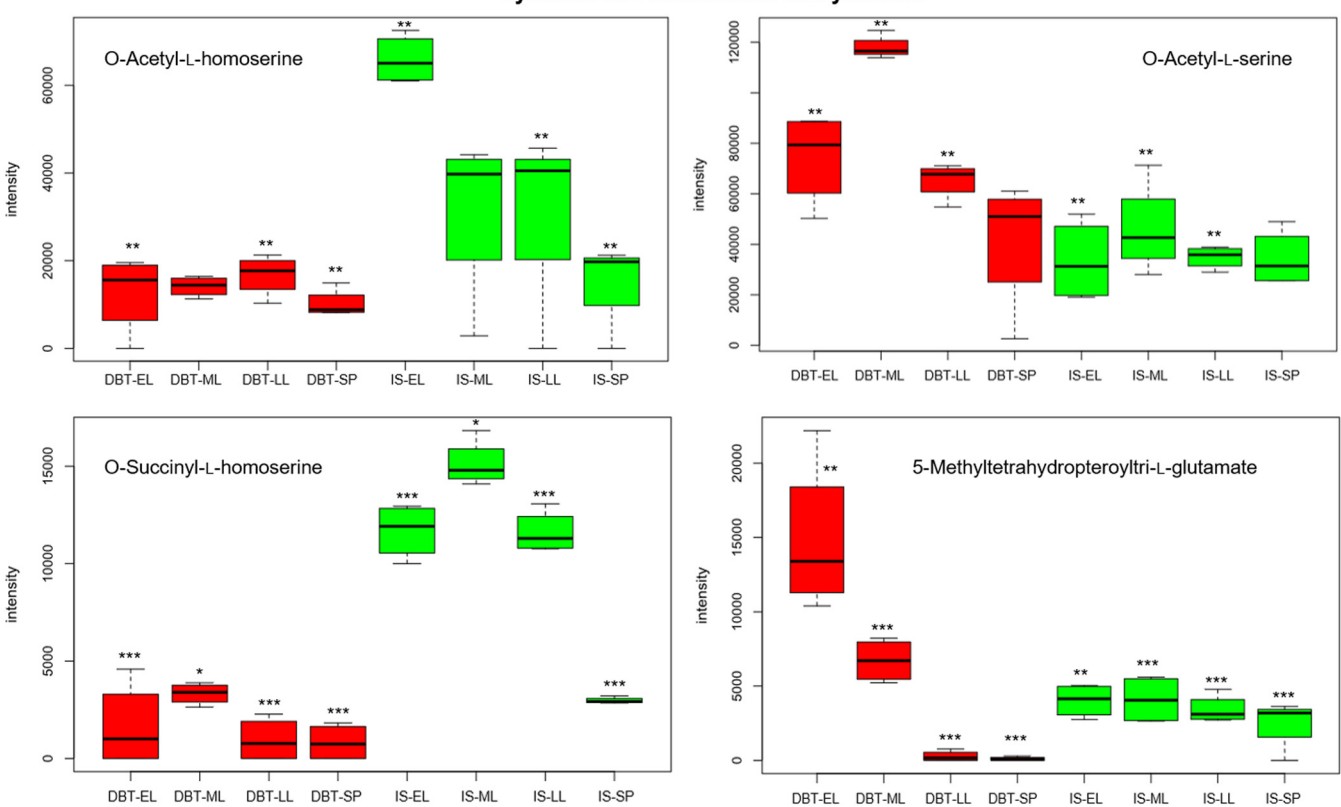

FIG 6 (A) Gene clusters of amino acid transport. (B) Proteins and metabolites of methionine and cysteine biosynthesis (see Table S1 for details of the abundance profiles and proposed functions of the proteins). Protein annotations are shown above the gene clusters, and gene names and IDs of the first and last genes are shown below the gene clusters. The growth phases are abbreviated as EL (early log), ML (mid-log), LL (late log), and SP (stationary phase). Bar charts represent the label-free quantification (LFQ) values showing the abundance profile of MetE in both the DBT (dibenzothiophene) and IS

source, inorganic sulfate, *R. qingshengii* IGTS8 appears to produce methionine using the cobalamin-dependent 5-methyltetrahydrofolate-homocysteine methyltransferase (MetH, IGTS8_peg1786). This enzyme was detected during all growth phases but could not be quantified (Fig. 2 and Table S1). The last step in methionine biosynthesis thus reflects the most conspicuous impact of sulfate availability on sulfur assimilation pathways. In addition to the cysteine and methionine biosynthesis proteins, the proteome revealed several proteins probably involved in cystine and methionine transport which were highly enriched in the dibenzothiophene culture (Fig. 6 and Table S1).

**Biosynthesis of the low-molecular weight thiols increased in the biodesulfurizing culture.** Enzymes catalyzing the biosynthesis of the actinobacterial low-molecular weight thiol, mycothiol, were slightly more abundant in the dibenzothiophene culture ($log_2$ fold change = 0.5 to 1.4), and their abundance increased toward the stationary phase (Fig. 7, Fig. S3A, and Table S1). Among these enzymes, MshA (IGTS8_peg5948), a glycosyltransferase which catalyzes the initial reaction in mycothiol biosynthesis, was uniquely present in the biodesulfurizing culture during the stationary phase. The mycothiol biosynthetic intermediate 1-*O*-(2-acetaamido-2-deoxy-$\alpha$-D-glucopyranosyl)-D-*myo*-inositol was uniquely present in the sulfate culture during the late log phase, and its level decreased with time where it became more abundant in the dibenzothiophene culture during the stationary phase ($log_2$ fold change = 2.09). To the contrary, the level of the subsequent intermediate, 1-*O*-(2-amino-2-deoxy-$\alpha$-D-glucopyranosyl)-D-*myo*-inositol, was significantly higher in the dibenzothiophene culture during the early log ($log_2$ fold change = 2.14) and mid-log ($log_2$ fold change = 1.78) phases, while it declined afterwards. The final product, mycothiol, was significantly more abundant ($log_2$ fold change = $-1.37$ to $-1.97$) in the sulfate culture throughout the life span (Fig. 7, Fig. S3A, and Table S3). The relative abundance of mycothione, the oxidation product of mycothiol, was not significantly different between the dibenzothiophene and inorganic sulfate cultures. In addition to mycothiol biosynthetic enzymes, the IGTS8 proteome revealed enzymes that catalyze mycothiol-dependent reactions of detoxification (Table S1 and see supplemental material for details). The dibenzothiophene culture also produced a level of enzymes involved in the biosynthesis of ergothioneine, another low-molecular weight thiol, higher than that of the sulfate culture. Four of the ergothioneine biosynthetic enzymes, EgtABCD, are encoded in an operon (Fig. 8), and their abundance consistently increased with the incubation time ($log_2$ fold change = 1.2 to 2.3) in the biodesulfurizing culture (Fig. 8, Fig. S3B, and Table S1). However, we could not identify in the IGTS8 genome a homolog of *egtE*, a pyridoxal phosphate-dependent $\beta$-lyase, which encodes the last enzyme of ergothioneine biosynthesis.

Although ergothioneine was not detected in the metabolome, we found a metabolite annotated as $\gamma$-glutamylcysteine, the first intermediate in the ergothioneine biosynthetic pathway (Fig. S3B and Table S3). Another interesting finding concerning ergothioneine biosynthesis is the presence in the dibenzothiophene culture of a significantly higher content of histidinol ($log_2$ fold change = 2.43) during the stationary phase (Fig. 8 and Table S3). Histidinol is a metabolite of biosynthesis of histidine, which is a precursor of ergothioneine.

**Metabolism of S-adenosylmethionine changed with the type of the sulfur source.** The dibenzothiophene culture had a statistically significantly higher level of S-adenosylmethionine synthetase (MetK, IGTS8_peg1600), though the $log_2$ fold change was less than 2.0 (Fig. 2 and Table S1). Under biodesulfurization conditions, 5′-deoxyadenosine, a metabolite of S-adenosylmethionine, was up to 4.3-fold more abundant compared to that in the inorganic sulfate culture. However, the abundance pattern

**FIG 6** Legend (Continued)
(inorganic sulfate) cultures. Significance of the data is attested by a Welch moderated *t* test as follows: NS for $P > 0.05$, * for $P \leq 0.05$, ** for $P \leq 0.01$, *** for $P \leq 0.001$, **** for $P \leq 0.0001$, DS for a protein which was uniquely identified in the dibenzothiophene cultures but not detected in the sulfate cultures, D for a protein which was identified but not confidently quantified. Metabolomics data are shown as boxplots displaying the distribution for each growth phase with the minimum, maximum, and median values for the dibenzothiophene (DBT) and inorganic sulfate (IS) cultures. Significance of the data (*P* value [rank], Wilcoxon test) is indicated by asterisks: * for $P < 0.01$, ** for $P < 0.05$, *** for $P < 0.1$, no asterisk for $P > 0.1$.

## Mycothiol biosynthesis

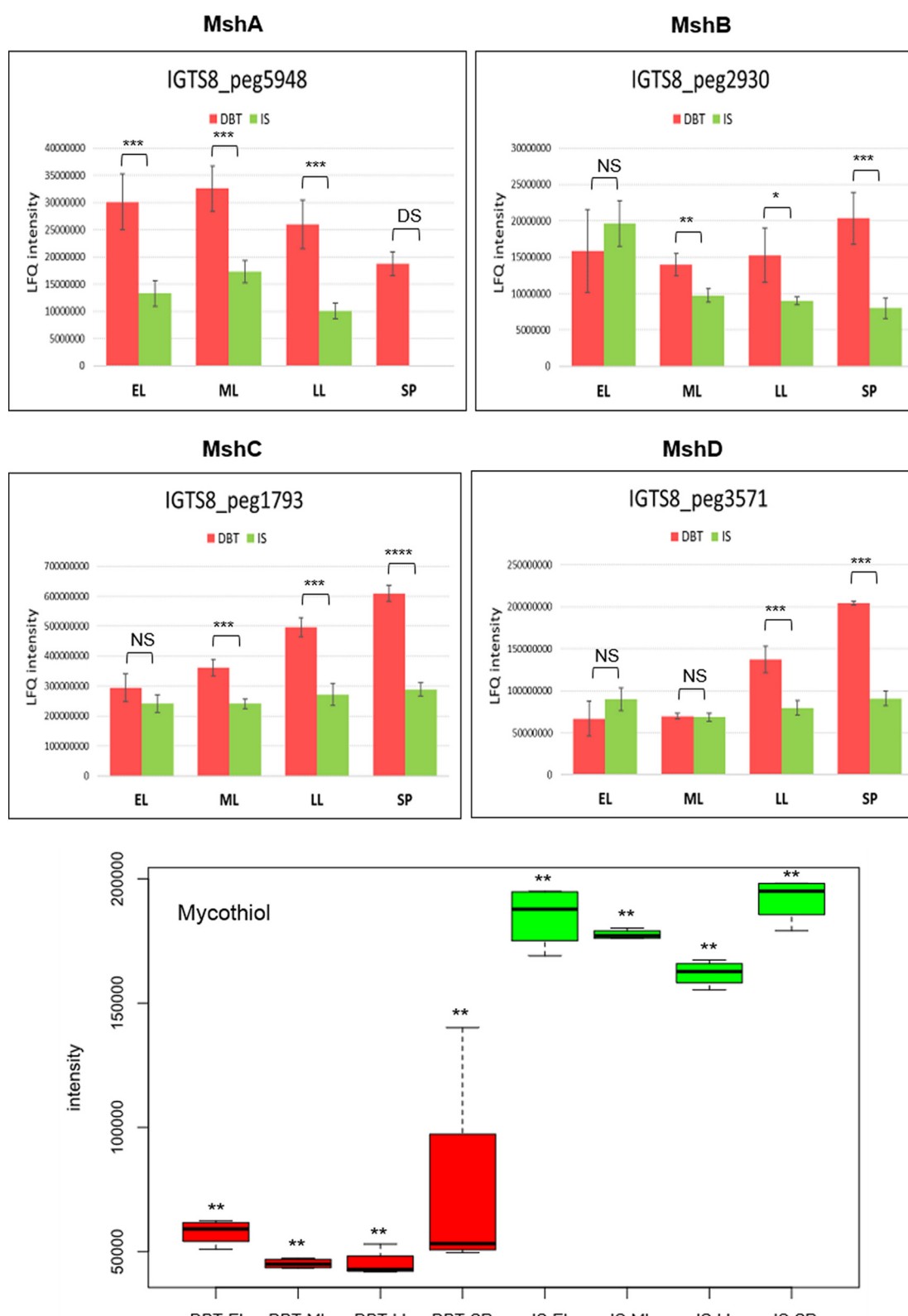

**FIG 7** Proteins of mycothiol biosynthesis (see Table S1 for details of the abundance profiles and proposed functions of the proteins). The growth phases are abbreviated as EL (early log), ML (mid-log), LL (late log), and SP (stationary phase). Bar charts

was reversed as the cultures entered the stationary phase where the inorganic sulfate culture had a 2.6-fold higher content. We also detected S-adenosylhomocysteine in both the biodesulfurizing and sulfate cultures (Table S3). An intriguing finding is the presence of a metabolite annotated as 1-aminocyclopropane-1-carboxylic acid, a metabolite of ethylene biosynthesis from S-adenosylmethionine in plants, that was uniquely present in the dibenzothiophene culture throughout and was about 1,500-fold more abundant during the stationary phase (Fig. S4 and Table S3). The IGTS8 genome, however, does not encode homologs of the enzymes that produce (1-aminocyclopropane-1-carboxylic acid synthase, EC: 4.4.1.14), oxidize (1-aminocyclopropane-1-carboxylic acid oxidase, EC: 1.14.17.4), or catabolize (1-aminocyclopropane-1-carboxylic acid deaminase, EC: 3.5.99.7) this metabolite.

**Biodesulfurization is associated with sulfur-sparing response.** To test whether the IGTS8 strain implemented sulfur sparing as a response to the sulfate starvation challenge in the dibenzothiophene culture, we counted the number of cysteine and methionine residues in the sequences of selected differentially synthesized proteins (supplemental material). Furthermore, we calculated the total content (as percentage) of cysteine and methionine in those proteins (Table S4). The first indicator for sulfur sparing in the biodesulfurizing culture was obvious from the protein sequence of the methionine biosynthesis enzymes. The cobalamin-independent MetE, which was much more abundant under biodesulfurization conditions, has only 2 cysteines and 7 methionine residues, whereas the cobalamin-dependent MetH isoenzyme has 11 cysteine and 27 methionine residues. Moreover, the highly abundant (SfnG) and uniquely present sulfur acquisition enzymes in the dibenzothiophene culture (TauD, AtsA, and subunit 2 of the sulfate activation complex) either have no cysteine and methionine or have only one cysteine residue and a maximum of seven methionine residues. In the sulfate-starved biodesulfurizing culture, the significantly upregulated MetY (IGTS8_peg3888) has no cysteine and only one methionine residue. It was also interesting to see that the 4S pathway enzymes have no cysteine and five methionine residues (DszA), one cysteine and no methionine residues (DszB), or one cysteine and three methionine residues (DszC). To the contrary, proteins that were not significantly, or were only slightly, upregulated in the dibenzothiophene culture, such as CysK (4 cysteines and 5 methionines) sulfite reductase (7 cysteines and 5 methionines), APS reductase (5 cysteines and 2 methionines), cystathionine-gamma-lyase (3 cysteines and 5 methionines), and cystathionine-$\beta$-synthase (4 cysteines and 8 methionines), have a higher number of cysteine and/or methionine residues. Accordingly, it can be inferred that under sulfate starvation (biodesulfurization) conditions, the IGTS8 strain avoids or limits the synthesis of cysteine- and methionine-rich proteins and depends instead on isoenzymes or paralogs having no or reduced cysteine and methionine content.

**Transcriptional regulators of sulfur metabolism in the IGTS8 strain.** The genome of the IGTS8 strain does not encode homologs of the known global regulators of sulfur metabolism such as the LysR-type CysB and Cbl of Gram-negative bacteria or the LysR-type CmbR and the TetR-type McbR of some Gram-positive bacteria (29, 36). However, the proteome revealed many transcriptional regulators, some of which are related to sulfur metabolism and are divergently encoded in operons for transport and utilization of organosulfur compounds (Fig. 3). One potential candidate as a global regulator of sulfur metabolism in the IGTS8 strain is the product of IGTS8_peg394, annotated as predicted transcriptional regulator of sulfate adenylyltransferase Rrf2 family that was uniquely detected in the dibenzothiophene culture throughout its life span. This annotation suggests a regulatory role for this protein in sulfur metabolism that was backed by BLAST

**FIG 7** Legend (Continued)
represent the label-free quantification (LFQ) values showing the abundance profile of the proteins in both the DBT (dibenzothiophene) and IS (inorganic sulfate) cultures. Significance of the data is attested by a Welch moderated $t$ test as follows: NS for $P > 0.05$, * for $P \leq 0.05$, ** for $P \leq 0.01$, *** for $P \leq 0.001$, **** for $P \leq 0.0001$, DS for a protein which was uniquely identified in the dibenzothiophene cultures but not detected in the sulfate cultures, D for a protein which was identified but not confidently quantified. Metabolomics data are shown as boxplots displaying the distribution for each growth phase with the minimum, maximum, and median values for the dibenzothiophene (DBT) and inorganic sulfate (IS) cultures. Significance of the data ($P$ value [rank], Wilcoxon test) is indicated by asterisks: * for $P < 0.01$, ** for $P < 0.05$, *** for $P < 0.1$, no asterisk for $P > 0.1$.

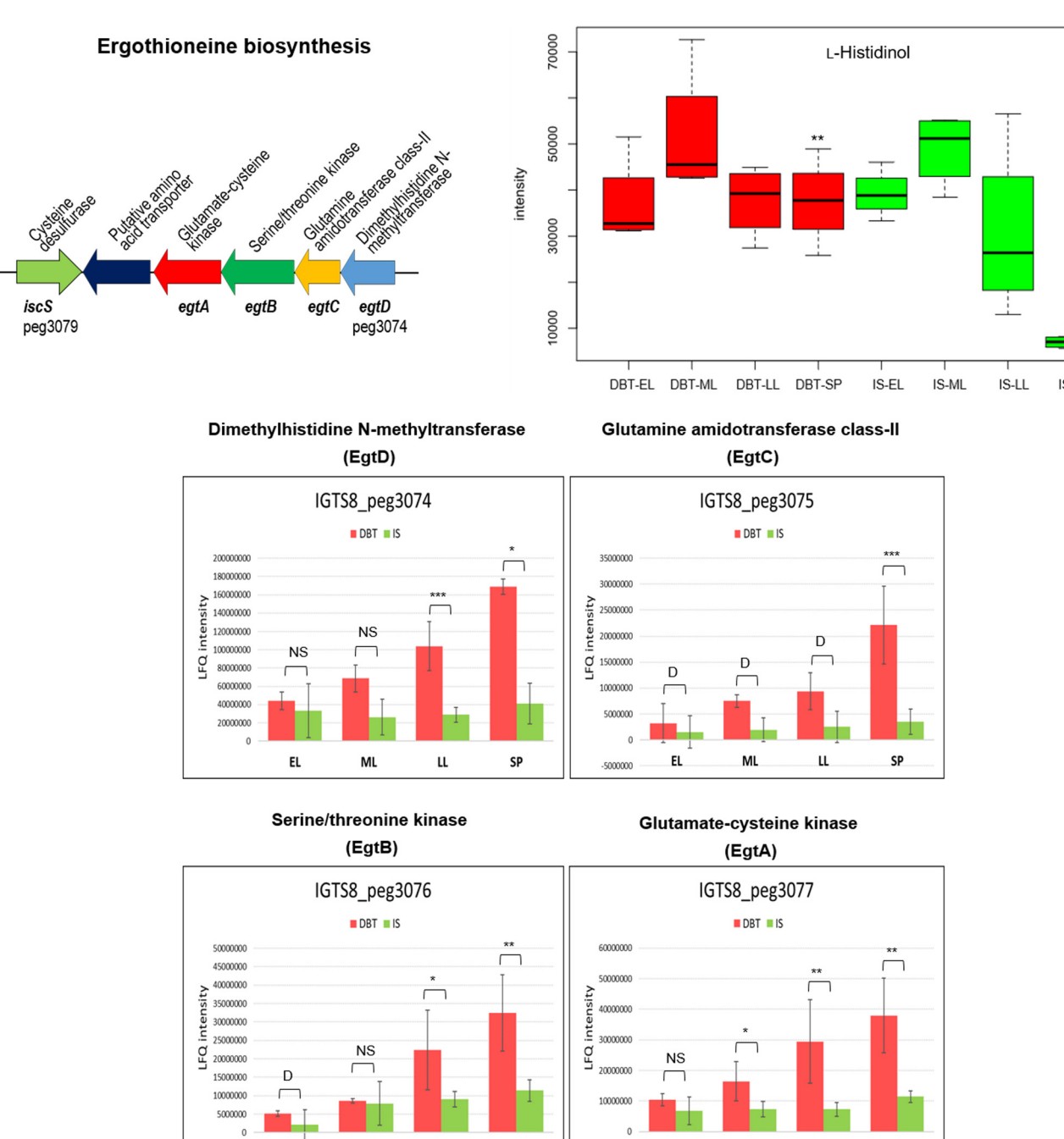

**FIG 8** Gene cluster, proteins, and a metabolite of ergothioneine biosynthesis (see Table S1 for details of the abundance profiles and proposed functions of the proteins). Protein annotations are shown above the gene clusters, and gene names and IDs of the first and last genes are shown below the gene clusters. The growth phases are abbreviated as EL (early log), ML (mid-log), LL (late log), and SP (stationary phase). Bar charts represent the label-free quantification (LFQ) values showing the abundance profile of the proteins in both the DBT (dibenzothiophene) and IS (inorganic sulfate) cultures. Significance of the data is attested by a Welch moderated $t$ test as follows: NS for $P > 0.05$, * for $P \le 0.05$, ** for $P \le 0.01$, *** for $P \le 0.001$, **** for $P \le 0.0001$, DS for a protein which was uniquely identified in the dibenzothiophene cultures but not detected in the sulfate cultures, D for a protein which was identified but not confidently quantified. Metabolomics data are shown as boxplots displaying the distribution for each growth phase with the minimum, maximum, and median values for the dibenzothiophene (DBT) and inorganic sulfate (IS) cultures. Significance of the data (P value [rank], Wilcoxon test) is indicated by asterisks: * for $P < 0.01$, ** for $P < 0.05$, *** for $P < 0.1$, no asterisk for $P > 0.1$.

search analysis showing 84% identity (E value: 2.2e−86) with a cysteine metabolism repressor CymR from *Rhodococcus* sp. strain AD45 (Table 1). Furthermore, the protein sequence alignment of IGTS8_peg394 with close homologues showed a high degree of conservation of amino acid residues, including the wHTH motif and dimerization

domain. The structural similarities were also confirmed by the three-dimensional (3D) model of CymR from *B. subtilis* strain 108 and *R. qingshengii* IGTS8 (Fig. S8). CymR is the master transcriptional regulator of cysteine metabolism in *Bacillus subtilis* (29, 38). The immediate genomic neighborhood, however, of IGTS8_peg394 does not encode any sulfur metabolism proteins. The dibenzothiophene culture also had 18- to 33-fold higher abundance of an ROK family transcriptional regulator (IGTS8_peg5940) that was even uniquely present in the dibenzothiophene culture during the stationary phase. This protein is divergently encoded downstream of an operon encoding a choline sulfatase (IGTS8_peg5938, AtsA1) and a putative $\alpha$-ketoglutarate-dependent dioxygenase (IGTS8_peg5939, TauD/TfdA family) that were likewise uniquely present in the dibenzothiophene culture proteome (Fig. 3 and Table S1). Among the interesting findings in the proteome of the biodesulfurizing culture is the upregulation (log$_2$ fold change = 2.0 to 2.5) of a putative extracytoplasmic function (ECF) sigma factor $\sigma^J$ (IGTS8_peg3414). ECF sigma factors constitute a large group of alternative sigma factors that play a role in the adaptive response of bacteria to environmental stimuli provoking cell envelope stress, oxidative stress, and virulence (39–41). Moreover, ECF sigma factors were implicated in the expression of sulfite-oxidizing enzymes for sulfite detoxification. The genomic location of $\sigma^J$ downstream of the *tau* gene cluster (presumably encoding a putative dibenzothiophene transporter) suggests a role in organosulfur transport and utilization under biodesulfurization (sulfate starvation) conditions.

## DISCUSSION

Providing dibenzothiophene instead of inorganic sulfate as the sole sulfur source to *R. qingshengii* IGTS8 was perceived as a sulfate starvation signal that elicited adaptive measures to overcome the hazardous consequences of the sulfate starvation stressor as also observed in *E. coli* and *Pseudomonas putida* (22, 42). *R. qingshengii* IGTS8 responded to the dibenzothiophene-imposed sulfate deficiency by reprograming sulfur metabolism via four main mechanisms, namely, (i) restricting sulfur assimilation, (ii) activating alternative sulfur assimilation pathways/enzymes, (iii) triggering sulfur-sparing response, and (iv) eliciting an oxidative stress protective machinery.

In some bacteria, including *E. coli*, *Bacillus subtilis*, *Mycobacterium tuberculosis*, *Pseudomonas aeruginosa*, and *Geobacillus thermoglucosidasius*, sulfate starvation leads to upregulation of the enzymes involved in sulfate activation/reduction (CysDN, CysH, CysI) and cysteine biosynthesis (CysK) (23, 26, 28, 36, 43–45). In contrast, our proteomics data did not reveal a significant change in the levels of these proteins and those of the reverse transsulfuration when *R. qingshengii* IGTS8 was forced to use dibenzothiophene as a sulfur source in the absence of sulfate. Therefore, it can be inferred that under biodesulfurization conditions *R. qingshengii* IGTS8 keeps sulfur assimilation and cysteine biosynthesis at minimum levels. This is in line with the observed sulfur-sparing response in the dibenzothiophene culture, where the most highly abundant proteins have contents of cysteine and methionine much lower than those of the low-abundance proteins (23, 42, 46).

It appears that cysteine biosynthesis in the dibenzothiophene culture increased slightly during the stationary phase. This might be necessary to make more cysteine available for the production of the redox buffers and detoxifying agents, mycothiol and ergothioneine, which is mandated by the stationary phase-associated nutritional and oxidative stress (47–50). The overproduction of enzymes of mycothiol and ergothioneine biosynthesis in the dibenzothiophene culture supports this conclusion. Mycothiol was probably much more involved in detoxification and redox-buffering reactions in the biodesulfurizing culture and, consequently, was detected in smaller amounts compared to that in the inorganic sulfate culture. This assumption is in line with the increased abundance of mycothiol-dependent detoxification enzymes in the dibenzothiophene culture toward the stationary phase. Alternatively, the stressed biodesulfurizing culture could utilize mycothiol as a source of biosynthetic precursors and energy (51), which might reduce its content compared to that of the sulfate culture.

The relatively higher abundance of *O*-acetyl-L-serine and cysteine synthases in the sulfate-starved dibenzothiophene culture suggests direct sulfhydrylation as the preferred route for cysteine biosynthesis (29, 36, 44), fully in line with an earlier *in silico* reconstructed model of sulfur metabolism in *Rhodococcus erythropolis* (52). A slightly higher abundance of cystathionine-$\beta$-synthase and cystathionine-$\gamma$-lyase in the sulfate culture points at a higher relevance of the reverse transsulfuration pathway for cysteine biosynthesis (32, 53). Our interpretations do not fully exclude the functionality of direct sulfhydrylation in the inorganic sulfate culture. We assume that it could be of higher relevance in the biodesulfurizing culture since using sulfide as the thiol group donor is metabolically more economic for the bacterial cell (37). The presence of more *O*-acetyl-L-homoserine sulfhydrylase (MetY) explains the low levels of *O*-acetyl-L-homoserine in the dibenzothiophene culture. MetY itself does not contain any cysteine residues, thus aligning with the sulfur-sparing response and providing further explanation for the preferential use of MetY and *O*-acetyl-L-homoserine for homocysteine biosynthesis under biodesulfurization conditions. To the contrary, the cysteine- and methionine-rich isoenzyme MetZ could be used preferentially for homocysteine production with *O*-succinyl-L-homoserine as the sulfide acceptor when *R. qingshengii* IGTS8 does not suffer from sulfur bioavailability problems in the presence of sulfate.

Reprogramming of methionine biosynthesis is one of the major findings of this study and represents a key switch in assimilatory sulfur metabolism in the IGTS8 strain. Although we cannot unambiguously explain why the methionine biosynthesis enzymes MetY and MetE were, in contrast to cysteine biosynthesis, boosted in the biodesulfurizing culture, we envisage that it reflects the much higher number of methionine than cysteine residues in the analyzed sulfur metabolism proteins. Furthermore, it pinpoints the energy-conscious smart adaptation of the IGTS8 strain manifested in the utilization of the cobalamin-independent methionine synthase (MetE), which spares at least part of the energy needed for cobalamin biosynthesis (44, 54) in the energy-stressed dibenzothiophene-grown cells due to the extra energy needed for dibenzothiophene utilization (4 mol of NADH per 1 mol of dibenzothiophene) (15, 52). With a similar rationale, following direct sulfhydrylation for methionine biosynthesis, with sulfide as the thiol source, would be less metabolically costly than going through transsulfuration using the metabolically more expensive cysteine as the thiol group donor (37). The increased dependence of the biodesulfurizing culture on MetE rather than MetH could be also an inherent component of the sulfur-sparing response of the IGTS8 strain (23, 42, 46). We base this conjecture on the fact that the turnover number of MetE is lower (~50-fold) than that of the cobalamin-dependent MetH (37, 55). Moreover, we showed in our sulfur-sparing analysis that MetE has a cysteine and methionine content much lower than that of its cobalamin-dependent counterpart MetH. The temporal decline of 5-methyltetrahydropteroyl-L-triglutamate, the methyl group donor for MetE, reflects higher consumption under the sulfate-limiting (biodesulfurization) conditions. In this context, our results are contradictory to an *in silico* model depicting sulfur metabolism in the biodesulfurizing *R. erythropolis* (52), which predicted methionine biosynthesis only by the cobalamin-dependent MetH.

In accordance with Aggarwal et al. (17), we propose direct CysI-catalyzed reduction of dibenzothiophene-derived sulfite to sulfide in *R. qingshengii* IGTS8, which is indispensable for assimilation and more energetically efficient than other indirect routes. The sulfate activation complex and CysH (APS reductase) probably work together to oxidize sulfite via an indirect sulfite oxidation pathway (56, 57). However, we propose that the primary task of the indirect sulfite oxidation route is to produce PAPS, not sulfate.

Oxidative formation of APS from sulfite and AMP with thioredoxin as an electron acceptor would be catalyzed by the *cysH*-encoded APS reductase. Once formed, APS could be phosphorylated to PAPS via the APS kinase domain fused to the sulfate activation complex, which may ensure direct substrate channeling for more effective catalysis as proposed for *M. tuberculosis* (31, 44). The affinity of the APS kinase for APS is several hundredfold greater than that of the ATP sulfurylase domain, and PAPS is formed

6-fold faster than ATP in the case of the *M. tuberculosis* enzyme (58). Thus, APS will kinetically partition almost exclusively toward PAPS synthesis.

The ATP-dependent phosphorylation of APS may facilitate sulfite oxidation by the thioredoxin-dependent APS reductase and overcome the very low reduction potential of thioredoxin ($-270$ mV) compared to that of the APS/sulfite plus AMP couple ($-60$ mV). The resulting PAPS is the universal sulfuryl group donor in reactions catalyzed by the sulfotransferase Stf0 (32, 59), which was also detected exclusively in the dibenzothiophene culture and encoded in a gene cluster downstream of the sulfate activation complex. The sulfotransferase may further drive the reaction in the direction of APS/PAPS formation. Going this way, the biodesulfurizing culture would benefit from the following: detoxification of sulfite (17, 56, 57), generation of ATP via partial oxidation of sulfite to sulfate (56), and satisfaction of the cells need for sulfated biomolecules (32, 59). This scenario explains why the biodesulfurizing culture overproduced the sulfate activation complex CysN2CD2 with the APS kinase fusion that is lacking in the CysN1D1 ATP sulfurylase (IGTS8_peg2385, 2386).

Although stimulation of the oxidative stress response in the dibenzothiophene culture, evident from the upregulation of alkylhydroperoxide reductase, agrees with previous studies (22, 60), the upshift in biosynthesis of the low-molecular weight thiols mycothiol and ergothioneine under biodesulfurization conditions has not been reported to date. Accordingly, we propose to extend the list of the sulfur starvation-induced proteins to include enzymes of low-molecular weight thiols biosynthesis. Mycothiol and ergothioneine are the actinobacterial analogs of the well-known redox buffer glutathione, and they play a crucial protective role under oxidative stress in addition to their detoxification capabilities (47, 49, 50, 61).

Manipulation of sulfur metabolism enzymes, other than those of the 4S pathway, for improving the biodesulfurization activity has been reported very rarely. Tanaka et al. (20) showed that disruption of the cystathionine-$\beta$-synthase gene by transposon mutagenesis in the biodesulfurizing *Rhodococcus erythropolis* KA2-5-1 led to a biodesulfurization activity in the presence of sulfate higher than that in the wild type and suggested that this phenotype is due to reduced cysteine biosynthesis in the mutant. Since the cystathionine-$\beta$-synthase mutant was grown on a mixture of 0.2 mM dibenzothiophene and 5 mM sodium sulfate, the culture would start using sulfate first as the sulfur source because it is the most preferred and also available in a sufficiently higher concentration than dibenzothiophene (52). In accordance with our proposed sulfur assimilation model, the reverse transsulfuration pathway should be activated to synthesize cysteine, but this will not work due to the disruption of a key enzyme, cystathionine-$\beta$-synthase. This condition might reduce the free cysteine pool, thus creating a sulfur starvation signal that could lead to a higher biodesulfurization activity.

Together with the naturally low sulfur requirements in bacteria, the sulfur-sparing response, elicited under biodesulfurization conditions, further restricts sulfur assimilation and, thus, represents a major barrier limiting the biodesulfurization activity (1, 18). To overcome this obstacle, Pan et al. (62) and Wang et al. (19) cloned a DNA fragment encoding a small peptide rich with methionine and cysteine in *R. qingshengii* and *Rhodococcus opacus* with the rationale to increase the cell's sulfur requirements and consequently enhance the biodesulfurization activity. Nonetheless, the sulfur-rich peptide failed to bring about remarkable increase in the biodesulfurization activity. Although the concept of increasing the cell's sulfur requirements *per se* is interesting, the approach should ensure that the "sulfur sink" is pivotal for the wellbeing of the biodesulfurizing cells. Presumably, the biodesulfurizing cultures did not need the sulfur-rich peptide designed by Pan et al. (62) and Wang et al. (19) and, therefore, did not express it to avoid waste of resources.

Considering those earlier studies and based on our proposed model for sulfur metabolism, we can now propose a hypothetical metabolic engineering scheme for future studies to improve the biodesulfurization activity. The rationale of our proposal is to force the biodesulfurizing cells to consume sulfur beyond their native limits (63) while

ensuring redox homeostasis. To be effective, a sulfur sink can be created by increasing the biosynthesis of sulfur-containing key metabolites that the cells might need under biodesulfurization conditions, such as the redox buffers mycothiol and ergothioneine. Another intervention would be to bypass the sulfur-sparing response of the sulfate-starved cells. One possible way is to knock out MetE, the low-turnover, sulfur-poor, and cobalamin-independent methionine synthase, to oblige the cells to induce, and rely on, the sulfur-rich and high-turnover paralog MetH. In parallel, we have to keep known signals of sulfate sufficiency/excess, such as sulfate, APS, sulfite, sulfide, and cysteine, at low levels (23, 36, 64). Maintaining reduced levels of sulfite, sulfide, and cysteine is also important to protect the biodesulfurizing cells from their toxicity at high concentrations (56, 65–67). In addition, at low intracellular concentrations, sulfide may stimulate respiration and ATP production (68).

**Conclusions.** Biodesulfurization is perceived as a sulfate limitation cue, which elicits a multifaceted adaptive response in the biodesulfurizing culture. The biodesulfurization phenotype is thus a reflection of the underlying alterations of not only the initial sulfur acquisition pathway but also assimilatory sulfur metabolism as a whole, in addition to the underlying oxidative stress. Together with the known low sulfur requirements, it appears that the biodesulfurization-induced sulfur sparing and overall constrained sulfur assimilation contribute to the prohibitively low biodesulfurization catalytic activity, which has been reported for decades. Accordingly, future endeavors in biodesulfurization research should dedicate efforts to metabolic engineering of sulfur metabolism to enable unprecedented improvements in the biodesulfurization efficiency, a prerequisite for the development of a commercially viable biodesulfurization technology. Genes encoding enzymes of the sulfate activation complex, potential uptake and efflux systems, and biosynthesis of methionine, cysteine, and low-molecular weight thiols are of particular interest.

## MATERIALS AND METHODS

**Composition of the chemically defined medium.** Sulfur-free chemically defined medium was prepared in deionized water with the following composition per liter: $KH_2PO_4$, 1.08 g; $K_2HPO_4$, 5.6 g; $NH_4Cl$, 0.54 g; $CaCl_2.2H_2O$, 0.044 g; $FeCl_2.4H_2O$, 1.5 mg; vitamins (cyanocobalamin 0.2 mg, pyridoxine-HCl 0.6 mg, thiamin-HCl 0.4 mg, nicotinic acid 0.4 mg, $p$-aminobenzoate 0.32 mg, biotin 0.04 mg, Ca-pantothenate 0.4 mg); trace elements ($ZnCl_2.7H_2O$ 70 μg, $MnCl_2.4H_2O$ 100 μg, $CuCl_2$ 20 μg, $CoCl_2.6H_2O$ 200 μg, $Na_2MoO_4.2H_2O$ 40 μg, $NiCl_2.6H_2O$ 20 μg, $H_3BO_3$ 20 μg). In the inorganic sulfate cultures, $MgSO_4.7H_2O$ (0.5 mM) was added as the sole sulfur source. In the dibenzothiophene cultures, dibenzothiophene (0.5 mM) was added (from a 100 mM stock in ethanol) as the sole sulfur source. In addition, $MgCl_2.6H_2O$ (0.5 mM) was added to the dibenzothiophene cultures to compensate for Mg concentration. No $MgCl_2$ was added to the inorganic sulfate cultures.

**Culturing conditions for proteomics and metabolomics studies.** To gain insights into the time-dependent biodesulfurization-driven physiological and metabolic adaptations, we conducted comparative and temporal systems biology studies (proteomics and metabolomics) on *R. qingshengii* IGTS8 (ATCC 53968) (8, 69). We chose the IGTS8 strain because it is the first and most extensively studied fuel-biodesulfurizing bacterium and it is frequently used as a model in biodesulfurization studies. We grew the IGTS8 strain in sulfur-free chemically defined medium under identical conditions in two cultures with the type of the sulfur source as the sole variable, and we compared the proteomes and metabolomes of both cultures during different growth phases. The cultures contained 20 mM glucose as the carbon source and a 0.5 mM concentration of either $MgSO_4$ (sulfur-sufficient condition) or dibenzothiophene (sulfur starvation condition) as the sole sulfur source. We selected dibenzothiophene because it is one of the most common organosulfur compounds in diesel and the model sulfur source for biodesulfurization studies. Four biological replicates were prepared and there was a separate set of cultures for each biomass harvesting time point, i.e., there was a total number of 16 cultures for each sulfur source (4 time points times 4 replicates). The cultures were inoculated from the respective starter cultures with 1% (vol/vol) resulting in a biomass load of 0.02 g/liter (wet weight). All cultures were incubated at 30°C in an orbital shaker (180 rpm). Growth was monitored by measuring the culture optical density at 600 nm ($OD_{600}$) at various time points, and cells were harvested at different growth phases. For the dibenzothiophene cultures, cells were harvested after 32 h (early log phase), 45.5 h (mid-log phase), 54.5 h (late log phase), and 67.5 h (stationary phase). For the sulfate-containing cultures, cells were harvested after 29 h (early log phase), 36 h (mid-log phase), 41 h (late log phase), and 45.5 h (stationary phase). The cells were harvested by centrifugation at 22.500 × $g$ for 15 min (4°C) in a Sorval Lynx 6000 centrifuge (Thermo Scientific, USA), and cell pellets were washed once in 50 ml of ice-cold K-phosphate buffer (0.1 M, pH 7) and collected by centrifugation at 30,000 × $g$ for 15 min. All harvesting steps were performed on ice in autoclaved centrifuge tubes, and washed cell pellets were stored at −80°C. All culture samples were subjected to the proteomics and metabolomics analyses.

**Quantitative proteomics. Sample preparation.** About 100 mg of the cell pellets were resuspended in Laemmli type buffer (10 mM Tris [pH 6.8], 1 mM EDTA, 5% $\beta$-mercaptoethanol, 5% SDS, 10% glycerol,

1/100 protease inhibitor cocktail [Merck P8340, Darmstadt, Germany]) at 10% wt/vol. The samples were vortexed and centrifuged at 1,000 × $g$ for 10 min. Protein concentration of all supernatants was determined using the RC DC protein assay (reducing agent and detergent compatible assay, Bio-Rad) according to the manufacturer's instructions in triplicate, and a standard curve was established using bovine serum albumin (BSA). For each sample, 20 $\mu$g of protein lysate was heated at 95°C for 5 min and stacked in an in-house prepared 5% acrylamide SDS-PAGE stacking gel at 50 V. Proteins in the gel were fixed with 50% ethanol/3% phosphoric acid, washed, and colored with Silver Blue. Gel bands were cut, washed with ammonium hydrogen carbonate and acetonitrile, reduced with 10 mM dithiothreitol, and alkylated using 55 mM iodoacetamide prior to overnight digestion at 37°C using modified porcine trypsin (Promega, Madison, USA) with a final trypsin/protein ratio of 1/50. The generated peptides were extracted with 60% acetonitrile in 0.1% formic acid followed by a second extraction with 100% acetonitrile. Acetonitrile was evaporated under vacuum and the peptides were resuspended in 40 $\mu$l of $H_2O$ and 0.1% formic acid before nanoLC-MS/MS analysis.

**NanoLC-MS/MS analysis.** Nano LC-MS/MS analyses were performed on a nanoACQUITY Ultra-Performance LC system (Waters, Milford, MA) coupled to a Q-Exactive Plus Orbitrap mass spectrometer (ThermoFisher Scientific) equipped with a nanoelectrospray ion source. The solvent system consisted of 0.1% formic acid in water (solvent A) and 0.1% formic acid in acetonitrile (solvent B). Samples were loaded into a Symmetry C$_{18}$ precolumn (0.18 by 20 mm, 5 $\mu$m particle size; Waters) over 2 min in 1% solvent B at a flow rate of 5 $\mu$l/min followed by reverse-phase separation (ACQUITY UPLC BEH130 C18, 200 mm by 75 $\mu$m i.d., 1.7 $\mu$m particle size; Waters) using a linear gradient ranging from 1% to 35% of solvent B for 79 min at a flow rate of 450 nl/min. The mass spectrometer was operated in data-dependent acquisition mode by automatically switching between full MS and consecutive MS/MS acquisitions. Survey full scan MS spectra (mass range 300 to 1,800) were acquired in the Orbitrap at a resolution of 70,000 at 200 $m/z$ with an automatic gain control (AGC) fixed at $3.10^6$ and a maximal injection time set to 50 ms. The 10 most intense peptide ions in each survey scan with a charge state of ≥2 were selected for fragmentation. MS/MS spectra were acquired at a resolution of 17,500 at 200 $m/z$, with a fixed first mass at 100 $m/z$, AGC was set to $1.10^5$, and the maximal injection time was set to 100 ms. Peptides were fragmented by higher-energy collisional dissociation with a normalized collision energy set to 27. Peaks selected for fragmentation were automatically included in a dynamic exclusion list for 60 s. All samples were injected using a randomized injection sequence. A sample pool comprising equal amounts of all protein extracts was constituted and regularly injected during the course of the experiment, as an additional quality control (QC). To minimize carryover, a solvent blank injection was performed after each sample. Monitoring protein identification rates and coefficients of variation (CV) of this QC sample revealed very good stability of the system: 2,105 of the 2,228 identified proteins, namely, 94%, showed a CV value lower than 20% considering all 4 injections.

**Data interpretation and statistical analyses.** Raw MS data processing was performed using MaxQuant software v1.6.0.16 (70). Peak lists were searched against an in-house generated database from the sequencing of the IGTS8 genome (6,734 sequences) using the RAST pipeline (71). The annotated genome is accessible via FigShare (https://dx.doi.org/10.6084/m9.figshare.14547426). MaxQuant parameters were set as follows: MS tolerance set to 20 ppm for the first search and 5 ppm for the main search, MS/MS tolerance set to 40 ppm, maximum number of missed cleavages set to 2, carbamidomethyl (C) set as fixed modification, oxidation (M) and acetylation (protein N-term) set as variable modifications. False-discovery rates (FDR) were estimated based on the number of hits after searching a reverse database and were set to 1% at both peptide spectrum match and protein levels. Data normalization and protein quantification were performed using the LFQ (label-free quantification) option implemented in MaxQuant using a "minimal ratio count" of one. The "match between runs" option was enabled using a 2-min time window after retention time alignment. All other MaxQuant parameters were set as default. To be considered for differential analysis, proteins must be identified in at least three out of the four replicates in both dibenzothiophene and inorganic sulfate cultures. The imputation of missing values and differential data analysis were performed using the open-source ProStaR software (72). A Welch moderated $t$ test was applied on the data set to perform differential analysis. Proteins were considered differential with a $P$ value lower than 0.05 and a log$_2$ fold change higher than 2 or lower than −2 with a minimum of 5 unique peptides. Proteins uniquely identified in one culture condition with a minimum of five unique peptides were also kept in a separate excel file. A complete data set has been deposited to the ProteomeXchange Consortium via the PRIDE partner repository (73) with the data set identifier PXD021362. Data were analyzed with principal-component analysis (PCA) on R (version 3.6.3) using the MetaboAnalyst package. The PCA was built considering the different proteins as individuals and the growth phases as variables. The PCA results indicate that the first two axes represent 94.6% of the data set inertia (variability of the data cloud).

**Metabolomics analyses. Sample preparation.** Cell pellets (30 mg) were dried under vacuum (Speed-Vac, ThermoScientific), and cell disruption was performed by grinding using a mortal with liquid nitrogen. A modified Bligh and Dyer protocol (74) was followed for metabolite extraction where 200 $\mu$l of hexane were added to the water/methanol/chloroform mixture followed by vortexing for 10 s, after which phase separation occurred as described by Bligh and Dyer. Internal standards d6 cholesterol (CDM Isotopes ref D-3373) and D-ABA (Olchemim ref 034 2721) as described in reference 75 were added. Nonpolar metabolome was collected from the methanol/hexane/chloroform solution, and the polar metabolome was collected from the water/methanol liquid phase. Both extracts were dried under vacuum (Speed-Vac, ThermoScientific) and stored at −80°C. For LC-MS/MS analysis, nonpolar metabolome samples were diluted 10 times with methanol before injection. The polar metabolome samples were resuspended in 500 $\mu$l of methanol for LC-MS/MS analysis.

**Untargeted LC-MS/MS analysis.** An Ultimate 3000 UHPLC (Thermo) coupled to an Impact II (Bruker) high-resolution quadrupole-time of flight (QTOF) was used to investigate the metabolome of the different samples. The metabolites were separated at 35°C on an Acquity UPLC BEH $C_{18}$ column (2.1 by 100 mm, 1.7 $\mu$m, Waters) coupled to an Acquity UPLC BEH $C_{18}$ precolumn (2.1 by 5 mm, 1.7 $\mu$m, Waters) using a gradient of solvent A (0.1% formic acid in water, Sigma-Aldrich) and solvent B (methanol-0.1% formic acid). The flow was set at 0.3 ml/min, starting with 5% of solvent B for 2 min and reaching 100% of solvent B from 10 to 13 min, and came back to 5% of B in 2 min for a total runtime of 15 min. The spectrometer was calibrated before the injections from 50 to 1,000 daltons (Da) using a mixture of 50 ml of isopropanol (Fisher Chemicals)/water (50/50, vol/vol), 500 $\mu$l of 1 M NaOH (Agilent Technologies), 75 $\mu$l of acetic acid, and 25 $\mu$l of formic acid (Sigma-Aldrich). The calibration mixture was injected at the beginning of each run for recalibration at the processing step. The data sets obtained were processed in Metaboscape 3.0 to investigate the nontargeted metabolome and perform the statistical analysis. Metabolite identification was made using analyte lists that were created from open access databases as described by Villette et al. (76). The following databases were interrogated: *E. coli* metabolome database (ECMDB, https://ecmdb.ca/), FooDB (https://foodb.ca/), Lipid Maps (https://www.lipidmaps.org/), Plant Cyc (https://plantcyc.org/), KNaspSAcK (http://www.knapsackfamily.com/KNApSAcK/), and Swiss Lipids (https://www.swisslipids.org/#/). Annotations were performed as described in reference 77, and metabolites were annotated to the level 3 of this classification. The metabolome coverage was estimated by dividing the number of putatively identified metabolites by the total number of *m/z* obtained.

**Statistical analyses.** Statistical analysis of the metabolomics data set was performed in Metaboscape 3.0 using the area of the peaks as the unit of reference. To comply with the small number of samples, a Wilcoxon rank sum test was performed for comparison of the samples by pairs (e.g., early log dibenzothiophene versus early log $MgSO_4$). A $\log_2$ fold change threshold of 1 and $-1$ was used to determine the differential metabolites, with a *P* value of $<0.05$. The annotated metabolites were also analyzed using PCA on R (4.0). The PCA was built considering the different metabolites as individuals and the growth phases as variables. The PCA results indicate that the first two axes represent 97.3% of the data set inertia (variability of the data cloud). Therefore, the main metabolites identified were placed in this PCA plot according to the two axes defined. Finally, the metabolites intensities were also graphically presented for each growth phase using boxplots drawn in the software R (4.0).

**Functional analyses of the detected proteins and metabolites.** Sulfur metabolism genes were identified in the genome of the IGTS8 strain using the Seed Viewer (78). Similarity search was performed using the BLAST program at the NCBI and UniProt databases with default settings. Mapping of the proteins and metabolites to metabolic pathways was performed with the KEGG mapper-annotate sequence by BlastKOALA (https://www.kegg.jp/kegg/tool/annotate_sequence.html) using the genus *Rhodococcus* as a data set and the MetaCyc databases (https://metacyc.org). The sequences of well-characterized CymR proteins from *Bacillus subtilis* strain 108 (UniProt accession number: O34527) and *Rhodococcus* sp. strain AD45 (UniProt accession number: A0A0D8I4J9) were obtained from the UniProt database (https://www.uniprot.org/) and aligned to IGTS8_peg394 using Clustal omega (79), followed by manual curation. The active domains including $\alpha$-helices and $\beta$-sheets were mapped from Shepard et al. (80). The folding and 3D structure of the proteins were analyzed using Phyre 2 (81).

**Sulfur-sparing analysis.** We checked the protein sequences of some of the most highly abundant and depleted proteins in the dibenzothiophene culture to look for indicators of sulfur-sparing response. This was done by counting the number of cysteine and methionine residues in the protein sequence (shown in the supplemental material). Moreover, we calculated the content of cysteine and methionine in each of those proteins as percentage of the total number of amino acids.

**Data availability.** A complete proteomics data set is available at the ProteomeXchange Consortium via the PRIDE partner repository with the data set identifier PXD021362.

## SUPPLEMENTAL MATERIAL

Supplemental material is available online only.

**SUPPLEMENTAL FILE 1**, PDF file, 1.5 MB.

**SUPPLEMENTAL FILE 2**, XLSX file, 0.01 MB.

**SUPPLEMENTAL FILE 3**, XLSX file, 0.02 MB.

## ACKNOWLEDGMENTS

This study is part of a research project funded by Kuwait Foundation for the Advancement of Sciences (grant no. P215-42SL-02). Proteomics experiments were supported by the French Proteomic Infrastructure (ProFI; ANR-10-INBS-08-03).

We declare no conflicts of interest.

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
