## [Reviewer comments · Microbiology Spectrum]

**Microbiology
Spectrum**

Biodesulfurization induces reprogramming of sulfur metabolism in *Rhodococcus qingshengii* IGTS8: Proteomics and untargeted metabolomics

Aur lie Hirschler, Christine Carapito, Lo c Maurer, Julie Zumsteg, Claire Villette, Dimitri Heintz, Christiane Dahl, Ashraf Al-Nayal, Vartul Sangal, Huda Mahmoud, Alain Van Dorsselaer, and Wael Ismail

Corresponding Author(s): Wael Ismail, Arabian Gulf University

Review Timeline:

Submission Date:

July 13, 2021

Accepted:

July 21, 2021

Editor: Jeffrey Gralnick

Reviewer(s): The reviewers have opted to remain anonymous.

Transaction Report:

DOI: <https://doi.org/10.1128/Spectrum.00692-21>

July 21, 2021

Prof. Wael Ismail
Arabian Gulf University
Life Sciences
Salmaniya
Manama
Bahrain

Re: Spectrum00692-21 (Biodesulfurization induces reprogramming of sulfur metabolism in *Rhodococcus qingshengii* IGT S8: Proteomics and untargeted metabolomics)

Dear Prof. Wael Ismail:

After consulting with another Editor and based on the responses and revisions to the prior round of review, your manuscript has been accepted, and I am forwarding it to the ASM Journals Department for publication. You will be notified when your proofs are ready to be viewed.

Sincerely,

Jeffrey Gralnick
Editor, Microbiology Spectrum

Journals Department
Table S2: Accept